# Maintenance of neuronal TDP-43 expression requires axonal lysosome transport

**Veronica H Ryan[1,2]\*, Sydney Lawton[1], Joel F Reyes[1], James Hawrot[1], Ashley M Frankenfield[3], Sahba Seddighi[1], Daniel M Ramos[2], Jacob Epstein[2], Faraz Faghri[2,4], Nicholas L Johnson[2,4], Jizhong Zou[5], Martin Kampmann[6], John Replogle[1], Yue Andy Qi[2], Hebao Yuan[1], Kory Johnson[1], Dragan Maric[1], Ling Hao[7], Mike A Nalls[2,4], Michael Emmerson Ward[1]\***

[1]National Institute of Neurological Disorders and Stroke, National Institutes of Health, Bethesda, United States; [2]Center for Alzheimer's and Related Dementias, National Institute on Aging, National Institutes of Health, Bethesda, United States; [3]Department of Chemistry, George Washington University, Washington, DC, United States; [4]DataTecnica, Washington, DC, United States; [5]National Heart, Lung, and Blood Institute, National Institutes of Health, Bethesda, United States; [6]Institute for Neurodegenerative Diseases, Weill Institute for Neurosciences, and Department of Biochemistry and Biophysics, University of California, San Francisco, San Francisco, United States; [7]Department of Chemistry and Biochemistry, University of Maryland, College Park, United States

**\*For correspondence:**
veronica.ryan@nih.gov (VHR);
michael.ward4@nih.gov
(MEmmersonW)

## eLife Assessment

In this **important** manuscript, Ryan et al perform a genome-wide CRISPR based screen to identify genes that modulate TDP-43 levels in neurons. They identify a number of genes and pathways and highlight the BORC complex, which is required for anterograde lysosome transport as one such regulator of TDP-43 protein levels. Overall, this is a **convincing** study, which opens the door for additional future investigations on the regulation of TDP-43.

**Abstract** TDP-43 mislocalization and pathology occurs across a range of neurodegenerative diseases, but the pathways that modulate TDP-43 in neurons are not well understood. We generated a Halo-TDP-43 knock-in human induced pluripotent stem cell (iPSC) line and performed a genome-wide CRISPR interference FACS-based screen to identify modifiers of TDP-43 levels in neurons. A meta-analysis of our screen and publicly available screens identified both specific hits and pathways present across multiple screens, the latter likely responsible for generic protein level maintenance. We identified BORC, a complex required for anterograde lysosome transport, as a specific modifier of TDP-43 protein, but not mRNA, levels in neurons. BORC loss led to longer half-life of TDP-43 and other proteins, suggesting lysosome location is required for proper protein turnover. As such, lysosome location and function are crucial for maintaining TDP-43 protein levels in neurons.

## Introduction

Mislocalization of transactive response DNA binding protein of 43 kDa (TDP-43) is a pathological hallmark of frontotemporal dementia and amyotrophic lateral sclerosis (FTD/ALS), two related

neurodegenerative diseases, and is a co-pathology in up to 57% of Alzheimer's disease cases (*Meneses et al., 2021*). Mislocalization of TDP-43 includes formation of aggregates in the cytosol and/or nucleus and nuclear clearance (*Tziortzouda et al., 2021*; *Baughn et al., 2023*). TDP-43 plays an important role in the regulation of gene expression and is critical for mRNA splicing. Nuclear loss of TDP-43 leads to increased alternative splicing events, a hallmark of TDP-43 proteinopathies (*Arnold et al., 2013*; *Highley, 2014*). TDP-43 also regulates its own expression at the mRNA levels by binding to its own 3' UTR, but this autoregulation is insufficient to maintain normal nuclear expression in disease (*Ling et al., 2013*; *Mackenzie and Neumann, 2017*; *Nana et al., 2019*). Whether other factors also regulate TDP-43 expression in neurons has not been systematically studied. Identification of such factors could reveal upstream disease mechanisms and potential therapeutic targets for neurodegenerative diseases.

The development of pooled CRISPR interference (CRISPRi) screens has allowed researchers to test the effect of thousands of genetic perturbations in parallel (*Doench, 2018*). CRISPRi screens are now possible in neurons derived from human induced pluripotent stem cells (iPSCs), enabling the evaluation of the effect of gene knockdowns on various phenotypes in these iPSC-derived neurons (iNeurons). Initial screens in iNeurons focused on survival-related phenotypes (*Tian et al., 2019*; *Tian et al., 2021*). More recently, FACS-based screens in iNeurons have enabled identification of modifiers of oxidative stress and peroxidized lipids (*Tian et al., 2021*). Historically, a practical challenge in the application of FACS-based screens to iNeurons has been that traditional genome-wide sgRNA libraries are large (>100k sgRNAs), making it difficult to sort sufficient cells to identify phenotypes. Therefore, sub-library screens have often been used in the past, limiting the scope of genes assayed. Recently, compact dual-sgRNA CRISPRi libraries have been developed (*Replogle et al., 2022a*), setting the stage for genome-wide FACS-based screens in iNeurons.

To uncover mechanisms leading to altered TDP-43 protein levels, we endogenously tagged the TARDBP gene with HaloTag in iPSCs, thereby generating a Halo-TDP-43 knock-in iPSC line to monitor TDP-43 expression in iNeurons. Using this line, we conducted a whole-genome CRISPRi FACS screen to identify modifiers of TDP-43 expression. Meta-analysis of this screen and other genome-wide CRISPRi FACS screens in iNeurons revealed common pathways shared across screens independent of the measured phenotype, enabling us to rank pathways specific for TDP-43 expression. All eight of the components of the BORC complex, which is necessary for axonal lysosome transport (*Pu et al., 2015*), were required to maintain neuronal TDP-43 expression. The genes encoding the eight subunits are BORCS1 (BLOC1S1), BORCS2 (BLOC1S2), BORCS3 (SNAPIN), BORCS4 (KXD1), BORCS5 (alias LOH1CR12), BORCS6 (alias C17orf59), BORCS7 (alias C10orf32), and BORCS8 (alias MEF2BNB). Knockdown of BORC reduced TDP-43 protein, but not RNA, levels. Surprisingly, metabolic labeling proteomics revealed that loss of BORC resulted in longer turnover time of TDP-43 protein. Notably, lysosome function and location also influenced Halo-TDP-43 levels in iNeurons. Our results identify BORC as a novel modifier of TDP-43 protein levels and point to a new mechanism by which TDP-43 levels are regulated in neurons.

## Results

### Generation of a Halo-TDP-43 knock-in iPSC line

The amount of TDP-43 expressed in neurons is tightly regulated, and TDP-43 undergoes autoregulation of its own transcript levels (*Avendaño-Vázquez et al., 2012*) to ensure levels are maintained at appropriate levels. To visualize and quantify the amount of TDP-43 in living cells, we used CRISPR technology to endogenously tag the N-terminus of TDP-43 with a HaloTag (*Figure 1A*). We selected a clone with correct editing based on sequencing. In this clone, the knock-in of the HaloTag was heterozygous, leaving the second allele of TDP-43 intact and unedited. The lead clone displayed a normal karyotype (*Figure 1—figure supplement 1A*), demonstrating that there were no major chromosomal abnormalities introduced during the gene editing or subsequent clonal expansion. Imaging of iNeurons differentiated from the knock-in iPSC line showed the majority of Halo signal in the nucleus, as is expected for TDP-43 localization (*Figure 1B*). A western blot against TDP-43 shows a shift of approximately 35 kDa in the two TDP-43 positive bands, corresponding to the molecular weight of Halo (33 kDa) (*Figure 1C*, *Figure 1—figure supplement 1B*). Knocking down TDP-43 using CRISPRi reduced the amount of both tagged and untagged TDP-43 compared to a non-targeting (NT) sgRNA

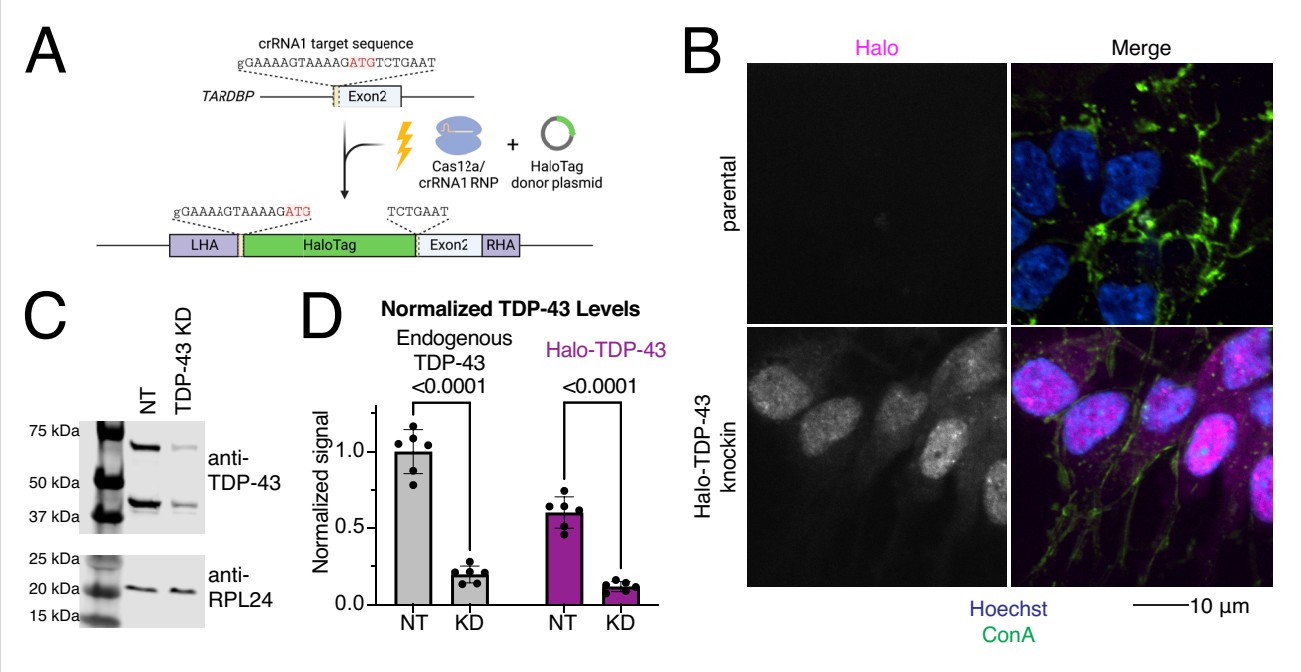

**Figure 1.** Generation of an endogenously HaloTagged TDP-43 induced pluripotent stem cell (iPSC) line. (**A**) Schematic of knock into the N-terminus of TDP-43. Cas12a-crRNA ribonucleoparticle (RNP) and HaloTag donor plasmid containing homology arms to TDP-43 were nucleofected into iPSCs. Halo was knocked into a single TDP-43 allele at the N-terminus of the protein, before exon 2, just after the start codon (ATG). (**B**) Microscopy validation of Halo-TDP-43 localization to nucleus. Halo in pink, ConA lipid dye in green, and Hoechst in blue. Halo signal is indicated in grayscale in left panels. Scale bar 10 μm. (**C**) TDP-43 western blot of non-targeting (NT) and TDP-43 KD neurons. Halo-TDP-43 is approximately 33 kDa heavier than untagged TDP-43, which corresponds to the molecular weight of Halo. RPL24 is shown as a loading control. TDP-43 KD decreases levels of both tagged and untagged TDP-43 compared to NT control sgRNA. (**D**) Quantification of western blot shown in (**C**) shows untagged (left) and HaloTagged (right) TDP-43 is decreased upon TDP-43 KD (n=6). Interestingly, Halo-TDP-43 is expressed at levels about one half as much as untagged TDP-43.

The online version of this article includes the following source data and figure supplement(s) for figure 1:

**Source data 1.** PDF file containing original western blot for *Figure 1C and D*, indicating the relevant bands and genotypes.

**Source data 2.** Original file for the western blot displayed in *Figure 1C*.

**Figure supplement 1.** i11w-hT induced pluripotent stem cells (iPSCs) have a normal karyotype.

**Figure supplement 1—source data 1.** PDF file containing original western blots for *Figure 1—figure supplement 1B*, indicating the relevant bands and genotypes.

**Figure supplement 1—source data 2.** Original files for western blots displayed in *Figure 1—figure supplement 1B*.

---

(*Figure 1C and D*). Expression of Halo-TDP-43 was roughly 60% of endogenous TDP-43, indicating that the Halo-tagged protein is either not transcribed or translated as efficiently as untagged protein, or the protein is degraded more readily.

## CRISPRi FACS screen identifies modifiers of Halo-TDP-43 levels in iPSC-derived neurons

To identify upstream modifiers that alter TDP-43 levels, we performed a genome-wide CRISPRi FACS-based screen in both Halo-TDP-43 iPSCs and iNeurons differentiated from our Halo-TDP-43 iPSC line using a dual sgRNA library, enabling 1000x coverage of the genome by sorting only 20 million iNeurons (*Replogle et al., 2022a*). We sorted for cells highly expressing the GFP sgRNA marker and then sorted the cells containing the lowest 25% of Halo signal into one tube and the cells containing the highest 25% of Halo signal into another (*Figure 2A*). sgRNAs targeting the TARDBP transcription start site decreased Halo-TDP-43 levels in both the iPSC and iNeuron screens (*Figure 2B* and *Figure 2—figure supplement 1A*, *Supplementary files 2 and 4*), indicating that our screen successfully identified genes that decrease TDP-43 levels. NT guides are clustered in the middle of the rank plot for each screen, showing that gene knockdown is needed to alter Halo-TDP-43 levels (*Figure 2B*

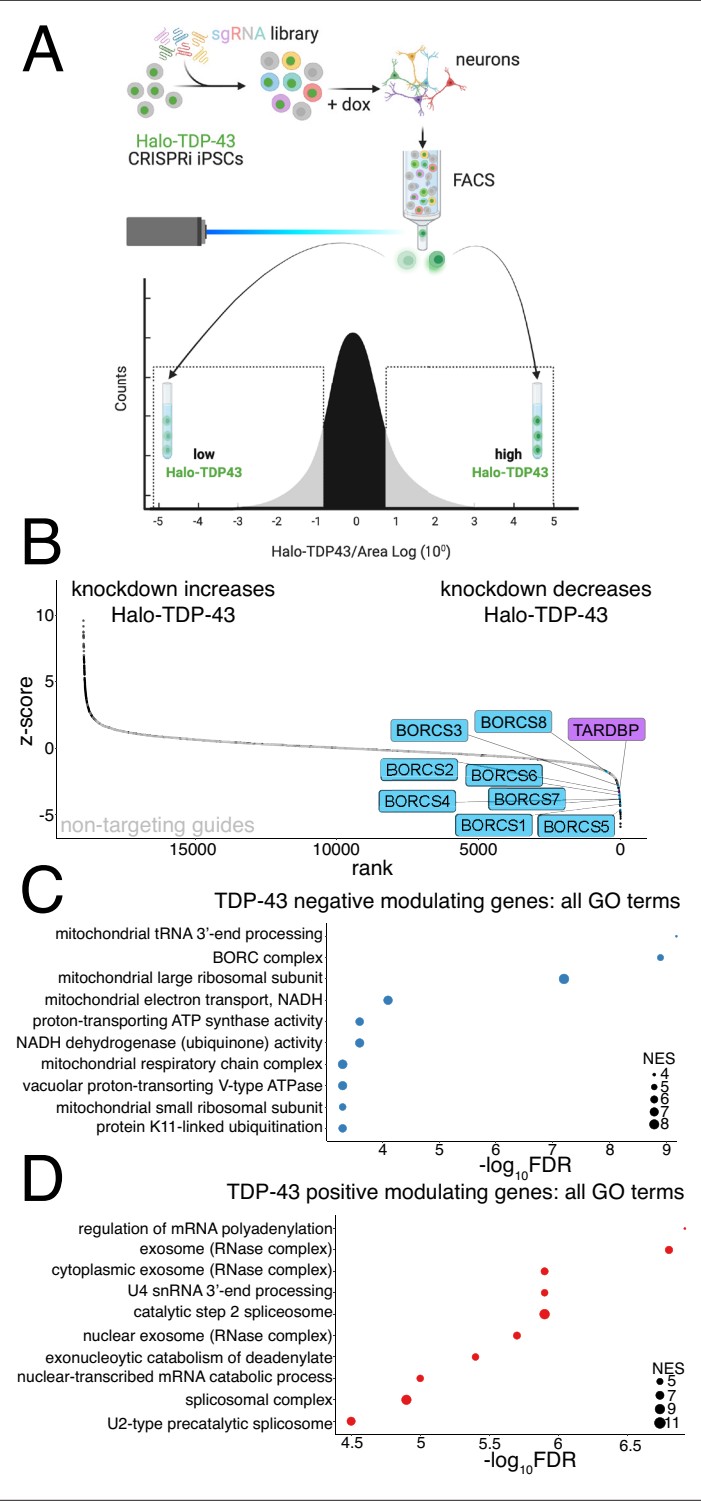

**Figure 2.** CRISPR interference (CRISPRi) screen identifies modifiers of TDP-43 protein levels in induced pluripotent stem cell (iPSC)-derived neurons. (**A**) Schematic of FACS screen. Halo-TDP-43 iPSCs were transduced with the dual guide library and selected. After selection, iPSCs were differentiated to neurons via Dox-inducible NGN2 expression. Neurons were FACS-sorted and populations expressing high Halo-TDP-43 and low Halo-TDP-43 were collected, DNA extracted, and sgRNA libraries sequenced. (**B**) Rank plot of screen results showing genes whose KD increases Halo-TDP-43 levels (left) and genes whose KD decreases Halo-TDP-43 levels, including TDP-43 itself (right). BORC genes are in blue. Non-targeting guides are indicated in gray and cluster in the middle, linear

*Figure 2 continued on next page*

*Figure 2 continued*

portion of the graph, demarking genes that do not change Halo-TDP-43 levels. (**C**) GO analysis of screen hits that reduce Halo-TDP-43 levels includes the BORC complex and many mitochondria-associated terms. The false discovery rate (FDR) is from the calculated permutation p-value of 1000 iterations. (**D**) GO analysis of screen hits that increase Halo-TDP-43 levels includes many hits related to mRNA processing. The FDR is from the calculated permutation p-value of 1000 iterations.

The online version of this article includes the following figure supplement(s) for figure 2:

**Figure supplement 1.** Induced pluripotent stem cell (iPSC) screen shows different hits than neuron screen.

and *Figure 2—figure supplement 1A*). Among genes whose knockdown decreases Halo-TDP-43 levels, we identified all eight components of the BORC complex, which regulates anterograde lysosome transport in axons as hits in the iNeuron, but not the iPSC screen (*Figure 2B and C* and *Figure 2—figure supplement 1A*, *Supplementary files 2 and 4*). Knockdown of genes related to mitochondrial function also decreases Halo-TDP-43 levels (*Figure 2C* and *Figure 2—figure supplement 1B*). Further, we identified genes whose knockdown increases Halo-TDP-43 levels, including genes involved in RNA processing (*Figure 2D* and *Figure 2—figure supplement 1C*, *Supplementary files 1 and 2*). In comparing the top GO terms associated with genes that increase or decrease Halo-TDP-43 levels, we found only one hypergeometric test (HGT) term associated with both the neuron and iPSC screens, nonsense-mediated decay independent of the exon junction complex, although it is associated with increased Halo-TDP-43 levels in iNeurons and decreased Halo-TDP-43 levels in iPSCs (*Figure 2—figure supplement 1D-G*). Thus, our hits are cell-type specific.

## Meta-analysis of open-access FACS CRISPRi screen data identifies common and novel fluorescence modifiers

CRISPRBrain (https://crisprbrain.org/) contains datasets from many CRISPRi and CRISPRa screens on different cell types examining various phenotypes. We identified three FACS-based CRISPRi screens on iNeurons that identify modifiers of reactive oxygen species (CellRox) and lipid peroxidation (Liperfluo) using fluorescent indicators (*Tian et al., 2021*) and tau aggregates (*Samelson et al., 2024*). We performed an unbiased fixed effects meta-analysis of these screens with our Halo-TDP-43 screen to determine if there were any common hits across screens. These common hits across multiple screens may be real hits that simultaneously change TDP-43 and alter ROS levels, lipid peroxidation, or tau aggregation; however, they may also be 'false-positives' that are altering general proteostasis, general cellular stresses, or neuron differentiation, where phenotypes examined in these screens are a secondary or indirect effect of knockdown. Interestingly, comparison of z-scores between screens shows many genes that either positively or negatively regulate both our screen and the CRISPRBrain screens (*Figure 3A–F*). With the increased power from the multiple screens, we were able to identify novel hits that were underpowered previously (*Figure 3A–F*). Genes were considered 'hits' if they were novel with directionality (either up or down) compared to the total population (the average of the hits in the CRISPRBrain screens). For the Halo-TDP-43 screens, genes were filtered by an LFC of 1 or –1 compared to the total population. We performed GO analysis on common hits from the meta-analysis across screens and found enrichment in many interesting categories, including ubiquitination, neddylation, autophagy, protein targeting, cell division/differentiation, and cytoskeletal-related terms (*Figure 3—figure supplement 1A–F*). On the other hand, some hits, like the BORC components, seem to be specific to our Halo-TDP-43 screen, so we chose to focus downstream analysis primarily on these hits that uniquely regulate TDP-43 biology. While this analysis provides a level of validation of our screen results, we picked genes from select categories to do a secondary, microscopy-based screen to further validate their effect on TDP-43 levels.

## Validation of screen hits narrows in on BORC as a modifier of TDP-43 levels

We cloned single sgRNA plasmids for 59 genes (*Supplementary file 5*) that either increased or decreased Halo-TDP-43 in iNeurons but not in iPSCs. sgRNAs were transduced into iPSCs individually, followed by differentiation of iPSCs into iNeurons. We used automated microscopy and analysis to visualize and quantify the Halo-TDP-43 signal in cells after each gene knockdown (*Figure 4A*). 34 of 59

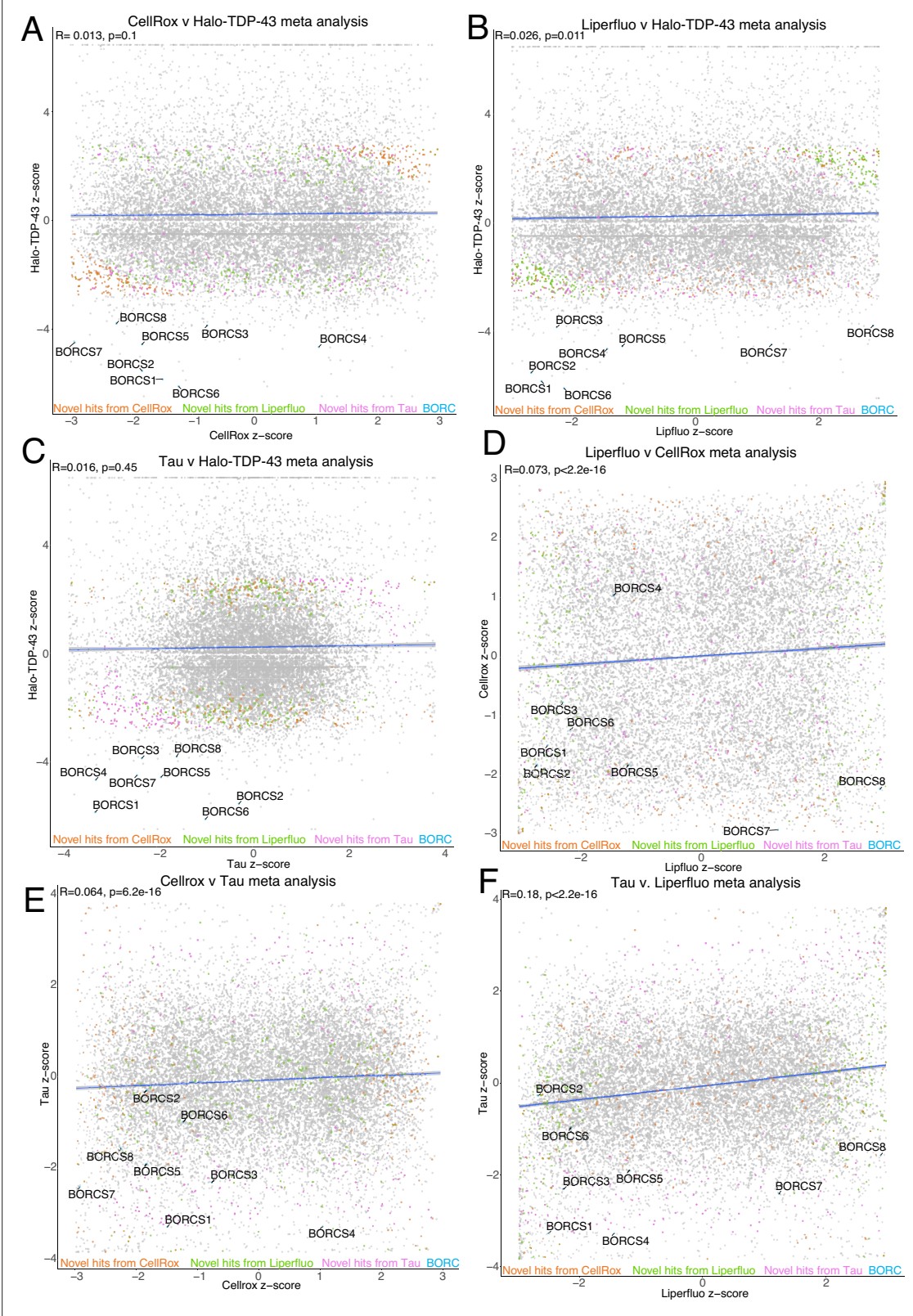

**Figure 3.** Meta-analysis of CRISPR interference (CRISPRi) screens identifies novel and common genes. (**A**) Comparison between CellRox and Halo-TDP-43 meta-analysis z-scores with novel hits highlighted. Correlation is indicated in blue with R and p-value at top left of graph. BORC genes are highlighted in blue, novel hits from Liperfluo, CellRox, and Tau screens as compared to our Halo-TDP-43 screen are indicated in green, orange, and pink, respectively. (**B**) Comparison between Liperfluo and Halo-TDP-43 meta-analysis z-scores with novel hits highlighted. Correlation is indicated in

*Figure 3 continued on next page*

*Figure 3 continued*

blue with R and p-value at top left of graph. BORC genes are highlighted in blue, novel hits from Liperfluo, CellRox, and Tau screens as compared to our Halo-TDP-43 screen are indicated in green, orange, and pink, respectively. Comparison between Tau and Halo-TDP-43 meta-analysis z-scores with novel hits highlighted. BORC genes are highlighted in light blue. Correlation is indicated in blue with R and p-value at top left of graph. (**C**) Comparison between Tau and Halo-TDP-43 meta-analysis z-scores with novel hits highlighted. Correlation is indicated in blue with R and p-value at top left of graph. BORC genes are highlighted in blue, novel hits from Liperfluo, CellRox, and Tau screens as compared to our Halo-TDP-43 screen are indicated in green, orange, and pink, respectively. (**D**) Comparison between Liperfluo and CellRox meta-analysis z-scores with novel hits highlighted. Correlation is indicated in blue with R and p-value at top left of graph. BORC genes are highlighted in blue, novel hits from Liperfluo, CellRox, and Tau screens as compared to our Halo-TDP-43 screen are indicated in green, orange, and pink, respectively. (**E**) Comparison between CellRox and Tau meta-analysis z-scores with novel hits highlighted. Correlation is indicated in blue with R and p-value at top left of graph. BORC genes are highlighted in blue, novel hits from Liperfluo, CellRox, and Tau screens as compared to our Halo-TDP-43 screen are indicated in green, orange, and pink, respectively. (**F**) Comparison between Tau and Liperfluo meta-analysis z-scores with novel hits highlighted. Correlation is indicated in blue with R and p-value at top left of graph. BORC genes are highlighted in blue, novel hits from Liperfluo, CellRox, and Tau screens as compared to our Halo-TDP-43 screen are indicated in green, orange, and pink, respectively.

The online version of this article includes the following figure supplement(s) for figure 3:

**Figure supplement 1.** Meta-analyses of Halo-TDP-43 and published CRISPR interference (CRISPRi) FACS screens.

(58%) tested genes altered Halo-TDP-43 levels in the microscopy validation (*Figure 4B–G*, *Figure 4—figure supplement 1A–D*, *Supplementary file 5*). For every gene tested except one, we observed a simple increase or decrease in Halo-TDP-43 levels without an obvious change in Halo-TDP-43 localization (*Figure 4B*). Since we used a single sgRNA plasmid for validation and not the dual guide plasmid used in the screen, some of the genes that did not pass this stage of validation may not have had sufficient gene knockdown in the validation to elicit the effect observed in the screen. However, we were interested primarily in the genes whose knockdown had the strongest effect on Halo-TDP-43 levels. To determine if results from Halo-TDP-43 expression assays also applied to endogenous, untagged TDP-43 levels, we selected 22 genes from different groups in our list that passed Halo validation and performed immunofluorescence microscopy for endogenous (untagged) TDP-43 (*Figures 4D–G and 5A and B*, *Figure 4—figure supplement 1E–F*, *Supplementary file 5*). Of these, 18 (82%) gene knockdowns showed changes in endogenous TDP-43 levels (*Figure 4D–G*, *Figure 4—figure supplement 1E–F*, *Supplementary file 5*). However, many gene KDs, particularly genes involved in ubiquitination and m6A RNA methylation, changed untagged TDP-43 levels in the opposite direction as they change Halo-TDP-43 levels (*Figure 4D–G*, *Supplementary file 5*). We posit that this Halo-untagged difference may be due to inherent destabilization of the HaloTagged TDP-43. KD of BORC subunit genes changed TDP-43 levels in the same direction for both Halo imaging and immunofluorescence (*Figure 5A and B*). To rule out the possibility of neighboring gene or off-target effects of CRISPRi, as has been reported previously (*Replogle et al., 2022b*), we examined the impact of BORC knockout (KO) on endogenous TDP-43 levels. Using the pLentiCRISPR system, which expresses the sgRNA of interest on the same plasmid as an active Cas9 (*Sanjana et al., 2014*), we found that KO of BORCS7 using two different sgRNAs decreased TDP-43 levels by immunofluorescence (*Figure 5*). Because of the known relationship between lysosomal dysfunction and FTD/ALS pathophysiology (*Root et al., 2021*; *Rhinn et al., 2022*; *Hung and Patani, 2024*; *Wallings et al., 2019*; *Talaia et al., 2024*; *Ling et al., 2019*; *Beckers et al., 2021*), we focused on the relationship between TDP-43 levels, lysosomal biology, and BORC.

## Maintenance of neuronal TDP-43 requires active and neuritically localized lysosomes

As lysosomes are the primary recycling center of the cell, breaking down macromolecules into their subunits, we first asked whether lysosome activity was required to maintain TDP-43 levels in iNeurons as has been observed in other cell types (*Ormeño et al., 2020*; *Leibiger et al., 2018*). We treated Halo-TDP-43 neurons with various drugs that disrupt distinct processes in the lysosome pathway and asked if Halo-TDP-43 levels changed. Chloroquine (decreases lysosomal acidity), CTSBI (inhibits cathepsin B protease), ammonium chloride ($NH_4Cl$, inhibits lysosome-phagosome fusion), and GPN (ruptures lysosomal membranes) all consistently decreased Halo-TDP-43 levels, while Bafilomycin A1 (BafA1, prevents lysosomal acidification and fusion with autophagosomes), LLOME (causes lysosomal membrane rupture), and apilimod (PIKfyve inhibitor) did not cause changes in Halo-TDP-43 levels

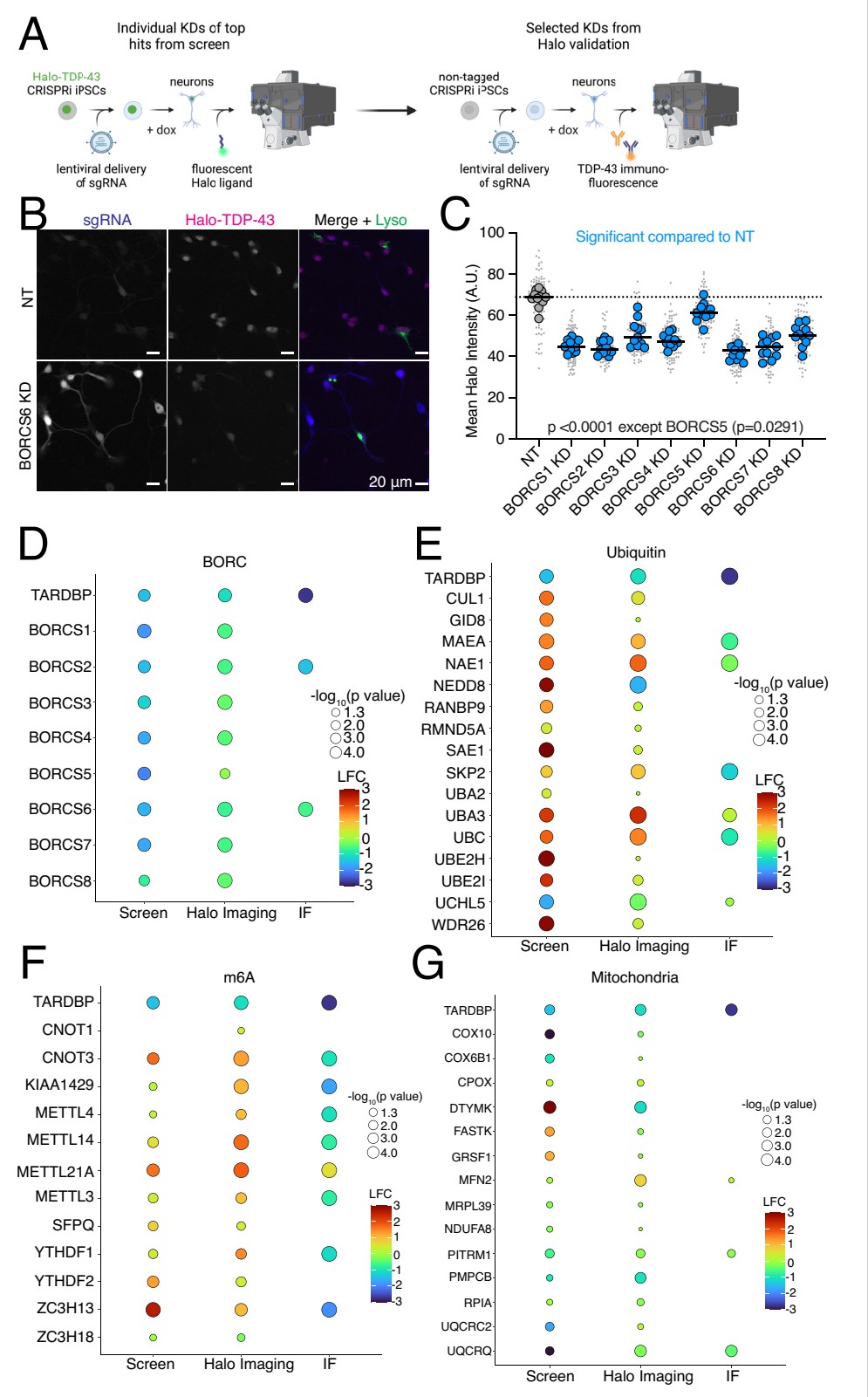

**Figure 4.** Validation of modulators of TDP-43 levels in induced pluripotent stem cell (iPSC)-derived neurons. (**A**) Schematic of hit validation. As a first validation, Halo-TDP-43 iPSCs were transduced with single sgRNAs against hit genes, and Halo levels were analyzed by microscopy, generating a list of Halo-validated hits. Then, some genes were selected from the Halo-validated hits, and untagged iPSCs were transduced with the same virus,

*Figure 4 continued on next page*

*Figure 4 continued*

and TDP-43 protein levels were assessed via TDP-43 immunofluorescence microscopy, identifying hits not affected by a HaloTag. (**B**) Representative images of Halo-TDP-43 live-cell imaging with BORCS6 KD. sgRNA plasmids contain a cytoplasmic BFP, enabling identification of cells expressing the sgRNA. For BORC genes, iPSCs were transduced with a lysosome marker (LAMP1-mApple) to ensure functional BORC KD through depletion of neuritic lysosomes. Scale bar represents 20 μm. (**C**) Quantification of BORC KD Halo-TDP-43 microscopy. All BORC genes tested (S1–S8) showed statistically significant decreases in Halo-TDP-43 levels compared to a non-targeting (NT) guide, indicated by blue dots. N=12 wells per genotype, 9 images per well (small gray dots). Significant p-values indicated on graph. (**D–G**) Summary graphs of screen, Halo live-cell imaging, and TDP-43 immunofluorescence imaging fold changes for (**D**) BORC KD. (**E**) Ubiquitin-associated gene KD. (**F**) m6A-associated gene KD, and (**G**) mitochondria gene KDs. Color indicates strength of $\log_2$fold change for each condition, circle size indicates $-\log_{10}$p-value; values below 1.3 are not significant, corresponding to a p-value of 0.05. No circle indicates the gene was not tested in that experiment (i.e. not all genes were tested by immunofluorescence).

The online version of this article includes the following figure supplement(s) for figure 4:

**Figure supplement 1.** Validation of neuron screen results shows many genes alter TDP-43 levels.

---

(***Figure 6A and B***, ***Figure 6—figure supplement 1A–C***). The varied mechanisms of action of these drugs indicated that many lysosomal perturbations could alter TDP-43 levels. Further, two protein translation inhibitors, anisomycin and cycloheximide (CHX), also decreased Halo-TDP-43 levels (***Figure 6A***, ***Figure 6—figure supplement 1B***).

Having shown that functional lysosomes are required to maintain Halo-TDP-43 levels, we next tested the role of lysosome location in iNeurons. As the BORC complex is required for anterograde movement of lysosomes within axons of neurons (***Pu et al., 2015***; ***Farías et al., 2017***), we first asked whether Halo-TDP-43 is transported on lysosomes. Our previous work showed that mRNA transport granules can be transported on lysosomes and that overexpressed TDP-43 co-trafficked with lysosome-associated transport granules (***Liao et al., 2019***). Indeed, in comparing motile Halo-TDP-43 with static Halo-TDP-43 in neurites, we found that motile TDP-43 was most often associated with lysosomes, not mitochondria (***Figure 6C and D***). Intriguingly, upon restricting lysosomes to the soma by knocking down a BORC subunit, we find that the fraction of motile Halo-TDP-43 in the periphery is reduced (***Figure 6E***, ***Figure 6—figure supplement 1D***).

## TDP-43 mRNA levels are unchanged upon BORC KD

We next asked whether TDP-43 RNA levels were also changed. Using qPCR, we found TDP-43 mRNA levels were unchanged or slightly increased upon BORCS6 KD (***Figure 7A***, ***Figure 7—figure supplement 1A***). As expected, we observed a statistically significant reduction of TDP-43 mRNA upon TDP-43 KD via CRISPRi (***Figure 7A***, ***Figure 7—figure supplement 1A***). We then performed total RNA sequencing of BORCS6 KD iNeurons. While BORCS6 mRNA levels were reduced by ~50% (***Figure 7B***), no change in TDP-43 mRNA levels occurred (***Figure 7C***), indicating that loss of BORC likely decreases TDP-43 protein levels independent of TDP-43 transcript expression.

As loss of BORC drastically changes the axonal transcriptome, particularly for mitochondrial genes (***De Pace et al., 2024a***), we wondered what the effect of BORC KD would be on the whole-cell transcriptome. We found that NT and BORCS6 KD samples were substantially separated on a PCA plot, with over 90% of the variability between the samples being explained by a single principal component (***Figure 7—figure supplement 1B***). We found 98 upregulated genes and 139 downregulated genes (***Figure 7D***) in BORCS6 KD neurons. GO analysis revealed an enrichment of terms related to neuron development and axon outgrowth (***Figure 7E***). As loss of TDP-43 can cause splicing changes and the inclusion of cryptic exons (***Mehta et al., 2023***; ***Gittings et al., 2023***; ***Cao et al., 2023***), we asked whether BORCS6 KD decreased TDP-43 levels sufficiently to induce inclusion of cryptic exons (***Brown et al., 2022***; ***Seddighi et al., 2024***). We did not observe an increase in expression of cryptic exons in genes that are commonly mis-spliced in iPSC-derived neurons after TDP-43 loss, including STMN2 and UNC13A. As such, BORCS6 KD does not reduce TDP-43 protein levels sufficiently to cause functional loss of splicing repression. As TDP-43 alternative polyadenylation (polyA) contributes to TDP-43 autoregulation and levels (***Avendaño-Vázquez et al., 2012***), we asked whether BORCS6 KD changed polyA site usage using REPAC (***Imada et al., 2023***). We found many sites changed upon BORCS6 KD (***Figure 7—figure supplement 1C***), with these genes relating to GO Biological Process terms like

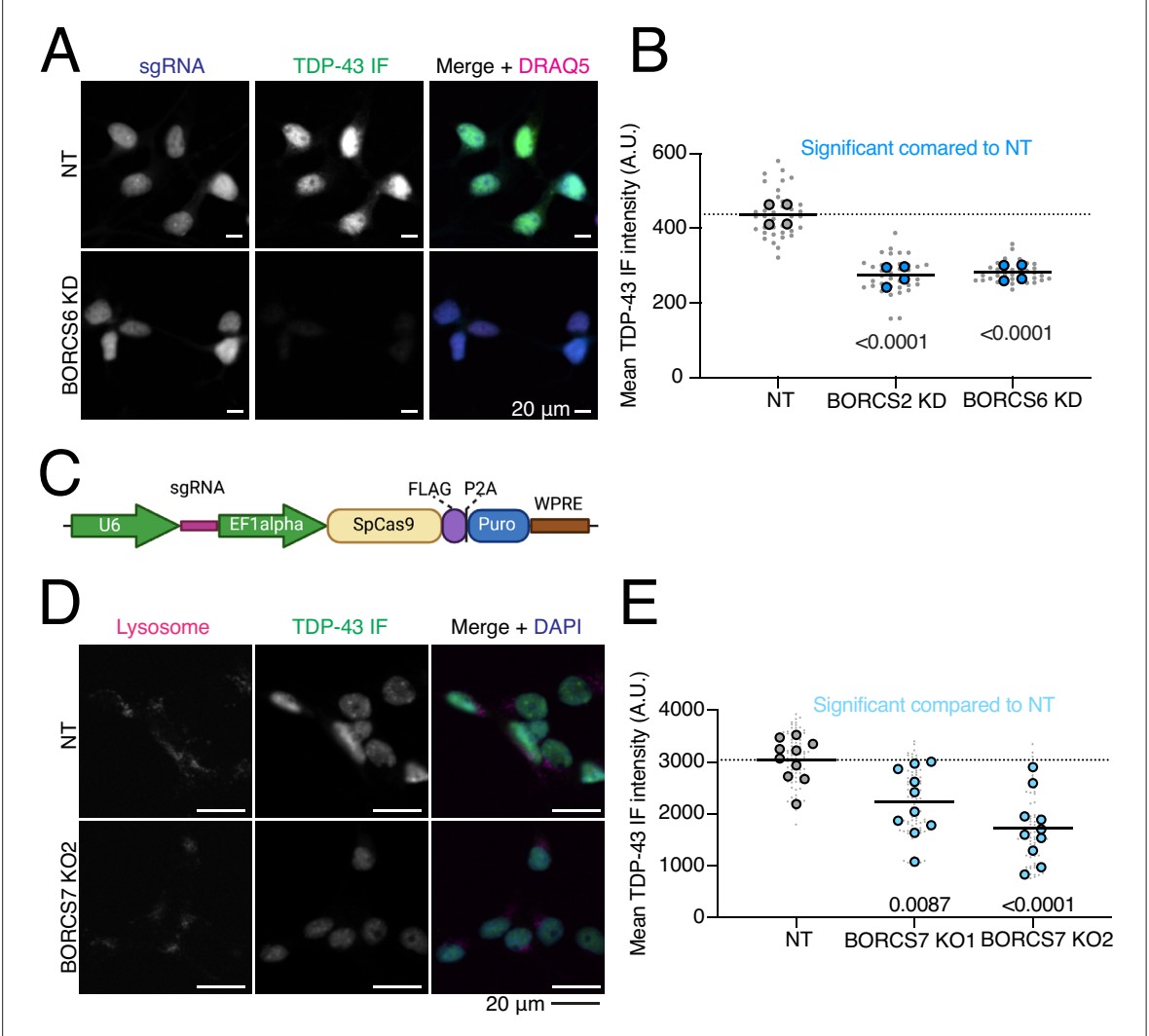

**Figure 5.** Endogenous validation of TDP-43 level modifiers in induced pluripotent stem cell (iPSC)-derived neurons. (**A**) Quantification of BORC KD TDP-43 immunofluorescence microscopy. Both BORC genes tested (S2 and S6) show significant decreases in untagged TDP-43 levels compared to a non-targeting (NT) guide, indicated by blue dots. N=4 wells per genotype, 9 images per well (small gray dots). Significant p-values indicated on graph. (**B**) Representative images of TDP-43 immunofluorescence imaging with BORCS6 KD. sgRNA plasmids contain a cytoplasmic BFP, enabling identification of cells expressing the sgRNA. Scale bar represents 20 µm. (**C**) Schematic of pLentiCRISPR plasmid. pLentiCRISPR enables lentiviral delivery of a plasmid expressing an sgRNA of interest under a U6 promoter and Cas9 under an Ef1alpha promoter. This enables knockout of a gene of interest targeted by an sgRNA. (**D**) Representative images of BORCS7 KO TDP-43 IF with lysosomes stained with anti-H4A3 antibody. Scale bar is 20 µm. (**E**) Quantification of TDP-43 immunofluorescence on BORCS7 knockout neurons. Both BORCS7 guides significantly reduced the amount of TDP-43 IF signal compared to an NT guide, light blue dots. N=10 wells, 9 images per well (light gray dots). Significant p-values indicated on graph.

splicing, catabolism, and transport/localization (*Figure 7—figure supplement 1D and E*). However, we did not observe any changes in TDP-43 polyA site usage upon BORCS6 KD (*Figure 7—figure supplement 1C*), so alternative polyA cannot explain the changes in TDP-43 protein levels.

## TDP-43, like the proteome, shows longer turnover time in BORC KD neurons

As loss of BORC prevents the transport of lysosomes to axons (*Farías et al., 2017*; *De Pace et al., 2024a*) and mRNAs can be transported on lysosomes to the periphery (*Liao et al., 2019*), we speculated that TDP-43 levels could be decreased in BORC KD iNeurons due to decreased axonal translation of TDP-43 mRNA. Using a TDP-43 mRNA N-terminally tagged with SunTag epitopes and with PP7 stem loops after the 3' UTR, we used microscopy to quantify the co-localization of TDP-43 new protein

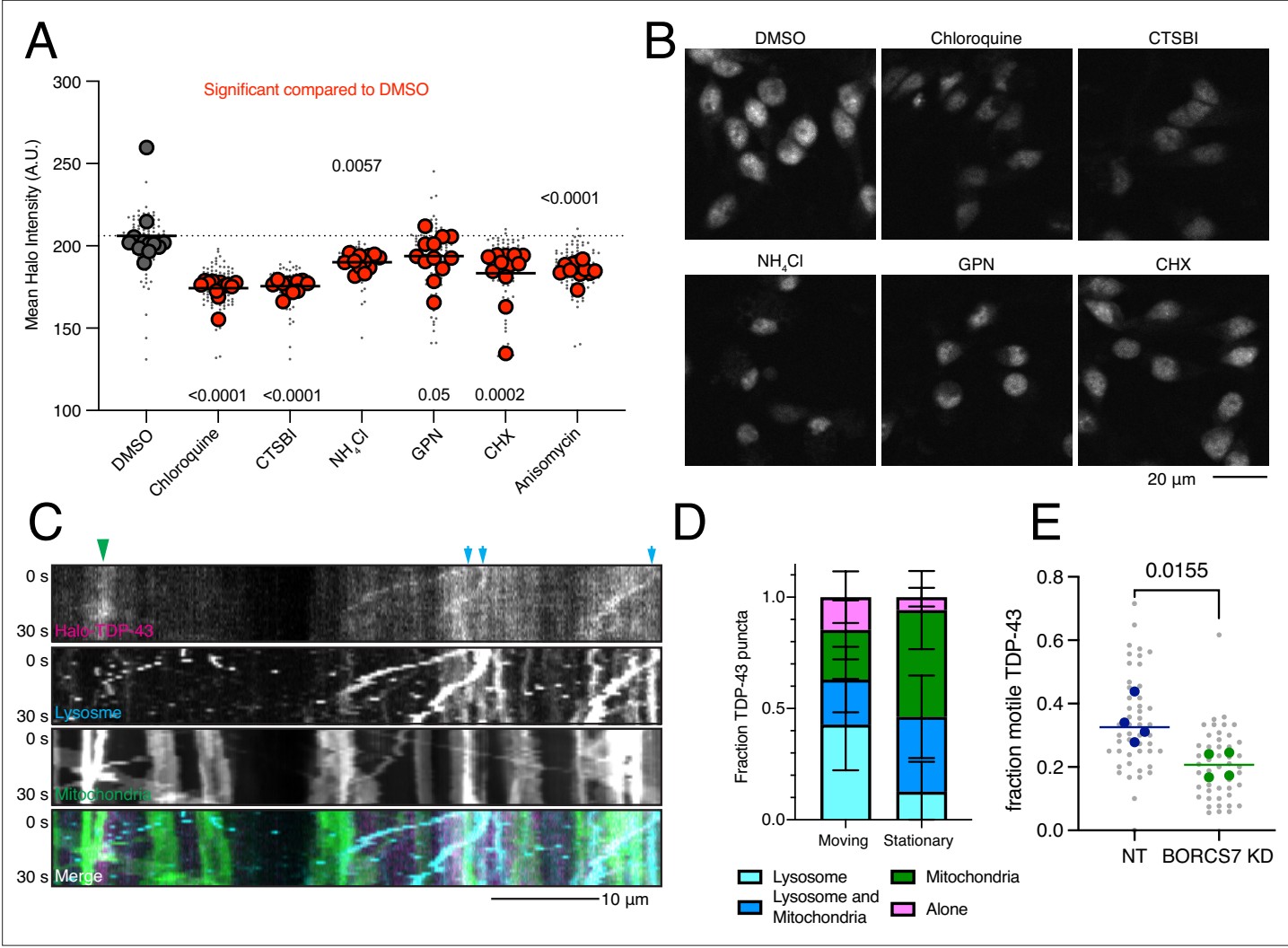

**Figure 6.** TDP-43 protein levels require active and neuritically localized lysosomes. (**A**) Drugs targeting different aspects of lysosome function decrease Halo-TDP-43 levels. All drugs tested were significant compared to a DMSO treatment, indicated by red dots. N=12 wells, 9 images per well (small gray dots), mean of each well is indicated by large dot. (**B**) Representative images of drug treatments that decrease Halo-TDP-43 levels. Scale bar is 20 μm. (**C**) Representative 30 s kymographs of Halo-TDP-43, lysosomes (LAMP1-mApple), mitochondria (mito-mEmerald), showing co-transport of Halo-TDP-43 with organelles. Green arrowheads indicate co-localization with static mitochondria; blue arrows indicate co-localization with motile lysosomes. Scale bar is 10 μm. (**D**) Quantification of Halo-TDP-43 signal with organelles. Motile Halo-TDP-43 in neurites is primarily co-transported with lysosomes or lysosomes and mitochondria together. Stationary Halo-TDP-43 is more likely to be associated with mitochondria. (**E**) Upon BORCS7 KD, TDP-43 is less mobile compared to a non-targeting (NT) guide, likely due to restriction of lysosomes to the soma. N=4 wells (mean per well indicated by large dot), 8 images per well (small dots).

The online version of this article includes the following figure supplement(s) for figure 6:

**Figure supplement 1.** Lysosome inhibitors alter Halo-TDP-43 levels.

synthesis (SunTag signal) with mRNA (PP7-coat protein signal) (***Ruijtenberg et al., 2018***). Surprisingly, we did not observe a change in TDP-43 translation (***Figure 8—figure supplement 1A***). As such, we hypothesized that loss of BORC might alter TDP-43 protein degradation rather than its synthesis. To test this hypothesis, we performed dynamic stable isotope labeling by amino acids in cell culture (dSILAC) to measure protein half-lives as well as whole neuron proteomics to measure protein abundances in BORCS6 KD and NT iNeurons. We found that TDP-43 total protein levels are decreased in BORCS6 KD neurons (***Figure 8A***), further validating our earlier microscopy findings. We observed a similar trend for many proteins in the proteome, with more proteins showing decreased abundance than increased (***Figure 8B***). Interestingly, TDP-43 shows longer half-life in BORCS6 KD neurons compared to the NT guide (***Figure 8C***), suggesting that protein degradation is potentially decreased in BORC KD iNeurons.

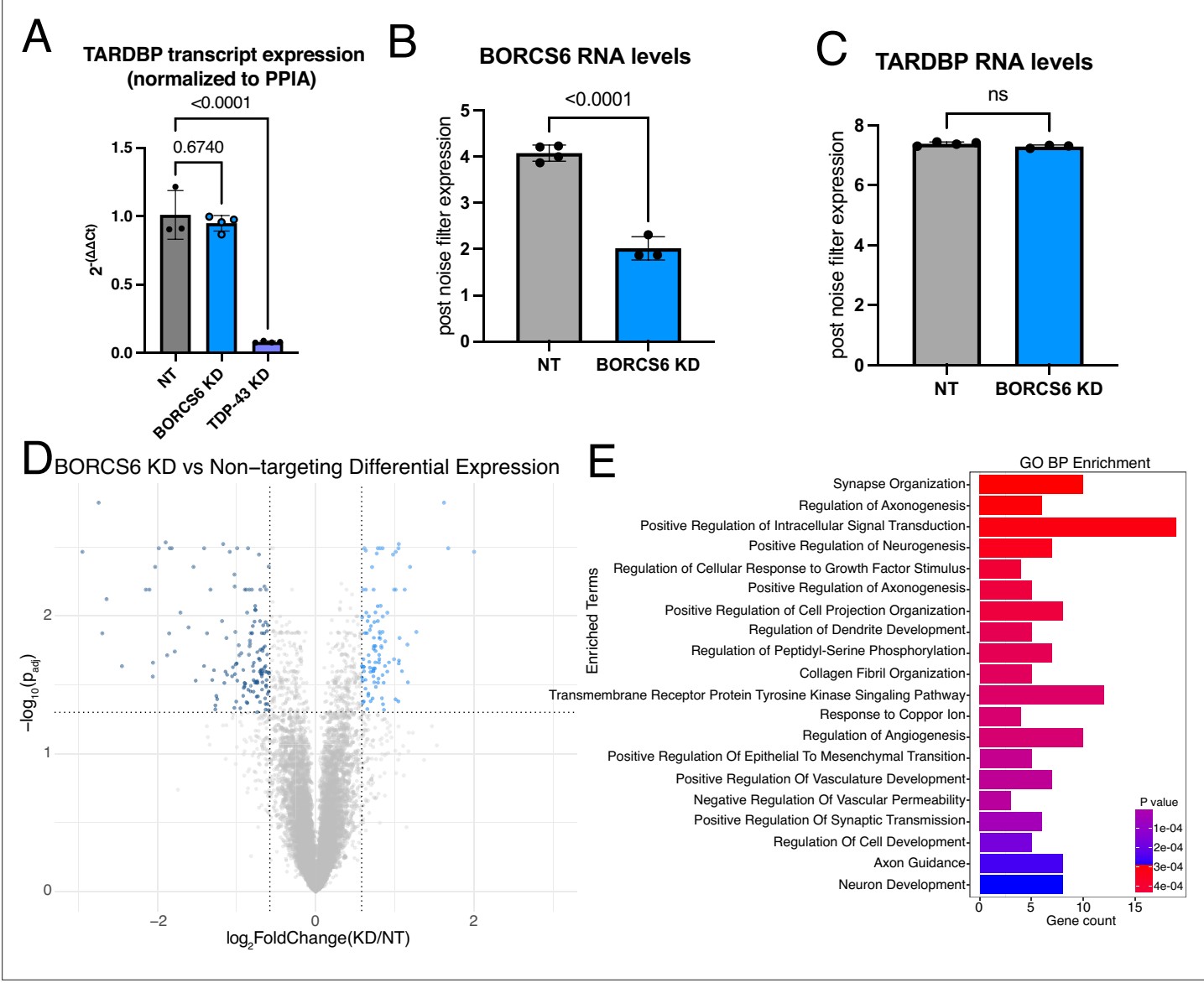

**Figure 7.** BORC KD does not affect TDP-43 RNA levels. (**A**) qPCR of non-targeting (NT), BORCS6 KD, and TDP-43 KD neurons showing decreased TDP-43 mRNA levels in TDP-43 KD, but not BORCS6 KD. Normalized to PPIA. NT n=3, BORCS6 KD n=4, and TDP-43 KD n=4. (**B**) Normalized transcript counts of NT and BORCS6 KD neurons showing about half as much BORCS6 mRNA in BORCS6 KD neurons. NT n=4, BORCS6 KD n=3. (**C**) Normalized transcript counts of NT and BORCS6 KD neurons showing no change in TARDBP (TDP-43) mRNA levels in BORCS6 KD neurons. NT n=4, BORCS6 KD n=3. (**D**) Volcano plot comparing BORCS6 KD with NT neurons. 98 genes increase expression upon BORCS6 KD (right of graph), while 139 genes decrease expression (left of graph). NT n=4, BORCS6 KD n=3. (**E**) GO Biological Process (BP) enrichment analysis from Enrichr of genes significantly altered in BORCS6 KD neurons compared to NT neurons shows several neuron and axon-related terms.

The online version of this article includes the following figure supplement(s) for figure 7:

**Figure supplement 1.** TDP-43 mRNA levels are unchanged after BORCS6 KD.

We observed similar results for the total proteome, with many proteins showing longer turnover at day 7 in BORCS6 KD neurons compared to NT neurons (*Figure 8D and E*). Longer half-lives were also observed in d15 neurons, although the changes in both abundance and half-life were reduced (*Figure 8—figure supplement 1B-E*), suggesting the neurons may be better able to compensate for the loss of BORC at later time points. In comparing the KEGG and REACTOME terms shown as enriched in our proteins with longer turnover times in BORCS6 KD neurons (*Figure 8F*, *Figure 8—figure supplement 1F*) to the mRNAs whose localization is altered in BORC KO neurons (*De Pace et al., 2024a*), we observe many of the same categories are enriched, including ribosome and biosynthetic pathways. We further queried our

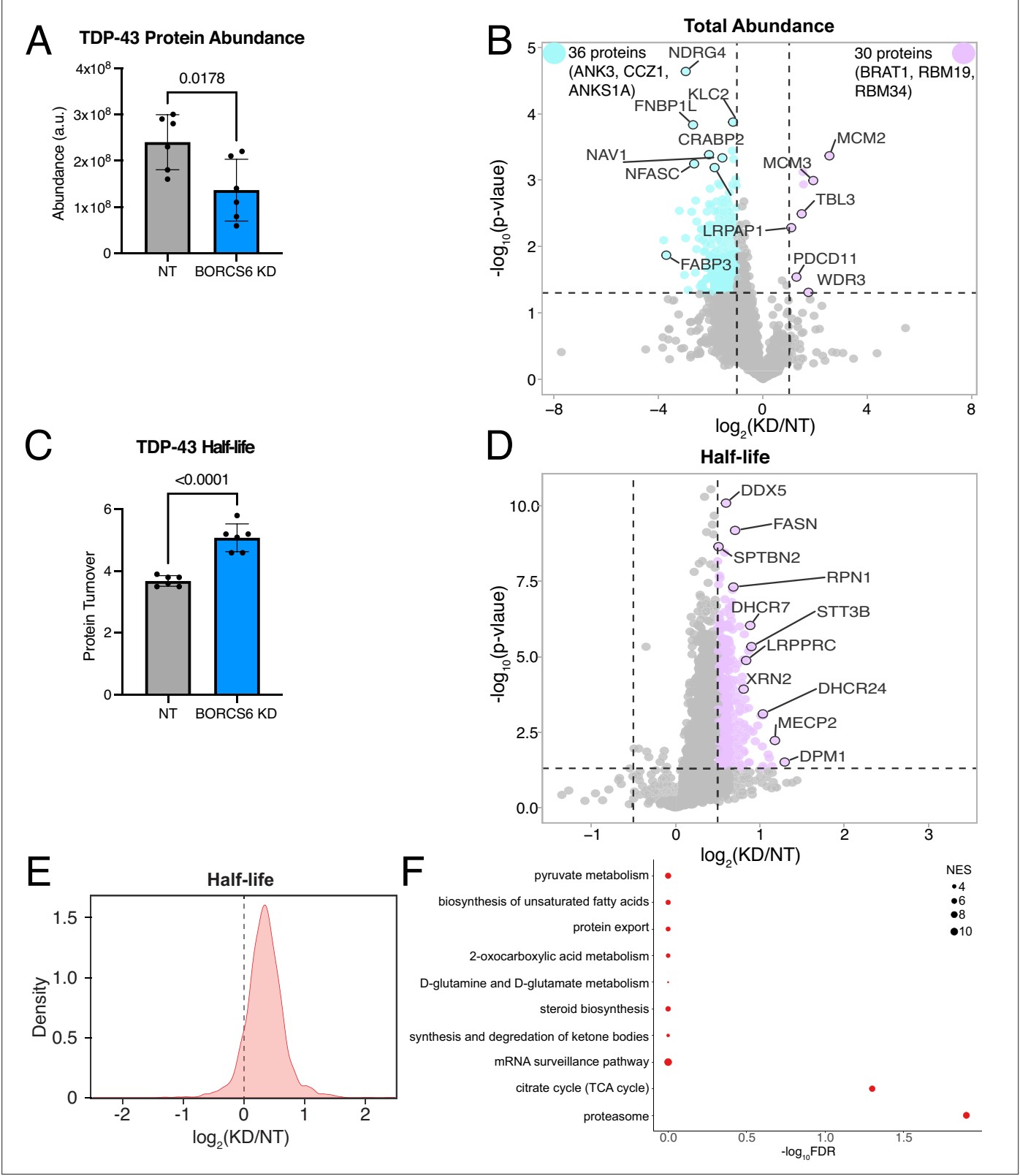

**Figure 8.** BORC KD alters the abundance and half-lives of TDP-43 and other proteins. (**A**) TDP-43 protein has decreased total abundance in day 7 BORCS6 neurons. (**B**) Volcano plot of total protein abundance comparing BORCS6 KD to a non-targeting guide. Some proteins (purple) have increased abundance, while a larger number (teal) have decreased abundance. Nonsignificant or low $\log_2$(fold change) proteins are indicated in gray. Vertical lines at $\log_2$(fold change) of 1 and −1, horizontal line at p-value = 0.05. (**C**) TDP-43 has longer half-life in day 7 BORCS6 KD neurons. (**D**) Volcano plot of

*Figure 8 continued on next page*

*Figure 8 continued*

half-life changes in BORCS6 KD neurons as compared to neurons expressing a non-targeting guide. Nonsignificant or low log$_2$(fold change) protein turnovers are indicated in gray. Horizontal line at p=0.05, vertical lines at log$_2$(fold change) of 0.5 or –0.5. (**E**) Density plot of global protein turnover shows longer protein half-lives upon BORC KD. (**F**) Gene ontology enrichment analysis using proteins with longer half-lives in BORCS6 KD neurons shows KEGG pathways related to protein export, proteasome, and metabolism. n=6.

The online version of this article includes the following figure supplement(s) for figure 8:

**Figure supplement 1.** Protein turnover is increased at d15 in BORC KD neurons.

data for posttranslational modifications (PTMs). We were able to identify 1333 phosphorylation sites. No PTMs were identified on TDP-43 at either time point, so PTM enrichment may be needed to query the role of PTMs in regulating the change in TDP-43 half-life with BORCS6 KD. We were able to detect 10 proteins with significantly changed phosphosites upon BORCS6 KD: KRI1 and RALY have phosphosites increased upon BORCS6 KD while OXSR1, ELAVL4, MAP4, ADD2, CSNK2B, FIP1L1, and RSF1 have phosphosites decreased upon BORCS6 KD at d7 (Figure S7G). At d15, only BASP1 showed a significantly decreased phosphosite (***Figure 8—figure supplement 1H***). Interestingly, ADD2, CSNK2B, MAP4, OXSR1, and RALY are detected to have differential phosphorylation and increased half-life, suggesting these phosphosites may contribute to regulation of the turnover of these proteins. As TDP-43 and the proteome show similar effects of BORCS6 KD on protein half-life, iNeurons are likely in steady state and making as much new heavy protein as they are degrading the light protein. As such, protein translation or protein degradation could be reduced. Although we did not observe a change in TDP-43 translation in our SunTag experiment, it may not be accurately reflecting the translation levels of TDP-43 due to the experimental necessity of overexpressing tagged TDP-43. As such, BORC may be affecting TDP-43 protein levels through a complicated mechanism.

## Discussion

CRISPRi screens are powerful tools to identify novel pathways regulating cellular functions. Here, we use a CRISPRi FACS screen to identify the lysosome transport complex, BORC, as a modifier of TDP-43 protein, but not RNA, levels. A screen in iPSCs confirms that BORC specifically acts on TDP-43 in iNeurons. Additionally, we made use of publicly available CRISPRi screen data to perform a meta-analysis. This meta-analysis identified common pathways across screens, enabling us to focus on TDP-43-specific hits. Common pathway hits across screens involve genes related to ubiquitination, neddylation, cell division/differentiation, and RNA modification. We think these pathways are related to protein level homeostasis in iNeurons, and our findings suggest a general utility of meta-analysis of novel screens in the context of published screens to prioritize hits for follow-up and identify phenotype-specific pathways.

KO of individual BORC subunits impairs lysosomal transport, leading to lysosomal depletion in axons and altered transport of transcripts that hitchhike on lysosomes to the periphery (***Pu et al., 2015***; ***Farías et al., 2017***; ***Liao et al., 2019***; ***De Pace et al., 2024a***; ***Snouwaert et al., 2018***; ***De Pace et al., 2020***). We initially hypothesized that loss of BORC would reduce neuritic translation of TDP-43. However, the lack of effect of BORC loss on TDP-43 mRNA or translation led us to focus on the role of lysosomes as a mechanism for how BORC affects TDP-43 protein levels. As KO of BORC subunits is known to alter lysosomal lipid metabolism, lysosomal content, size, and lead to decreased autophagic flux (***Yordanov et al., 2019***; ***Wu et al., 2023***; ***de Araujo et al., 2020***; ***Jia et al., 2017***), we asked whether BORC loss altered TDP-43 half-life. We found that BORC loss leads to longer half-lives of TDP-43 and many other proteins. Although TDP-43 is degraded by autophagy, and thus lysosomes (***Ormeño et al., 2020***; ***Leibiger et al., 2018***), this change in TDP-43 half-life due to lysosomal positioning has not been previously observed. As lysosomes can be active in axons (***Farfel-Becker et al., 2019***), this slower turnover of TDP-43 may imply that lysosomes degrade proteins in axons and are important for maintaining axonal homeostasis locally. Further work is needed to better understand this counterintuitive mechanism. TDP-43 protein levels are also decreased by lysosome-inhibiting drug treatments, further demonstrating the requirement of active and properly localized lysosomes for maintenance of TDP-43 levels. Interestingly, the decreased amount of TDP-43 in iNeurons upon BORC KD was insufficient to cause inclusion of cryptic exons which have been observed in models of TDP-43 loss of function (***Avendaño-Vázquez et al., 2012***; ***Mehta et al., 2023***; ***Gittings et al., 2023***; ***Brown et al., 2022***). It is possible that we did not observe cryptic exons simply because the levels of TDP-43 loss were insufficient to induce alternative splicing (in our hands,

TDP-43 KD via CRISPRi results in ~20% of TDP-43 remaining, while BORC KD results in ~60% of TDP-43 remaining in iNeurons). Alternatively, the longer half-life of TDP-43 due to BORC KD affects cryptic exon inclusion; more work is needed to examine this possibility.

Defects in lysosome function and intracellular transport are associated with FTD/ALS (*Root et al., 2021*; *Nicolas et al., 2018*). Our studies raise the possibility that lysosomal dysfunction is an upstream actor exacerbating TDP-43 loss-of-function phenotypes in disease. In this scenario, lysosomal mislocalization drives TDP-43 loss, which might further disrupt its autoregulation in disease. Interestingly, BORC KO in mice leads to neonatal death (*Snouwaert et al., 2018*; *De Pace et al., 2020*), and mutations in BORCS8 cause infantile neurodegeneration (*De Pace et al., 2024b*). While these neurodegeneration-causing BORC mutations reduce anterograde lysosome transport (*De Pace et al., 2024b*), TDP-43 levels have not been studied in patients with this disorder. Thus, we speculate that patients have altered TDP-43 levels, which could be sufficient to contribute to the observed neurodegeneration. As we were unable to detect cryptic exon formation in our BORC KD neurons, this potential TDP-43 loss of function in BORC patients may not lead to cryptic exon pathology observed in FTD/ALS patient tissue, so further work would be needed to understand the role of this intermediate TDP-43 loss in patients.

As TDP-43 pathology is associated with both loss- and gain-of-function effects (*Ling et al., 2013*; *Yang et al., 2014*; *Lee et al., 2011*), understanding the pathways that lead to altered TDP-43 levels is crucial to identifying effective therapeutic targets for neurodegeneration. At the same time, the more specific these pathways are for TDP-43, the less likely treatments targeting these pathways will cause unwanted off-target effects. We demonstrate that CRISPRi screens are a powerful method to identify these specific pathways. While we did not identify genes that specifically affected TDP-43 localization in iNeurons, the identified genes and pathways that specifically alter TDP-43 levels, like BORC subunits, will enable us to interrogate ways to change TDP-43 protein levels as potential therapies. Additionally, as more screens are performed and can be added to meta-analyses, we will gain further insight into nonspecific pathways affecting protein levels and thus are unlikely to be effective therapeutic targets to specifically hit targets. In conclusion, these results provide important insight into the pathways affecting TDP-43 levels in neurons.

## Methods

**Key resources table**

| Reagent type (species) or resource | Designation | Source or reference | Identifiers | Additional information |
|---|---|---|---|---|
| Cell line (*Homo sapiens*) | i11w-hT | This paper | | iPSC |
| Cell line (*Homo sapiens*) | i11w-mNC | *Tian et al., 2019*; 10.1016/j.neuron.2019.07.014 | | iPSC |
| Antibody | Anti-TDP-43 (Rabbit polyclonal) | Proteintech | 10782-2-AP, RRID:AB_615042 | 1:1000 IF, 1:1000 WB |
| Antibody | Anti-RPL24 (Rabbit polyclonal) | Proteintech | 17082-1-AP, RRID:AB_2181728 | 1:1000 WB |
| Antibody | Anti-Halo (mouse monoclonal) | Promega | G921A | |
| Other | Janelia Fluor 646 HaloTag Ligand | Promega | HT1060 | 200 µg/mL |
| Recombinant DNA reagent | MK-EF1a-LAMP1-SBP-mApple (plasmid) | *Snyder et al., 2022* 10.1002/alz.13915 | I76 | |
| Recombinant DNA reagent | MK-Ef1a-mito-mEmerald-WPRE (plasmid) | This paper | M33 | |
| Recombinant DNA reagent | sgRNA (plasmid) | *Tian et al., 2019*; 10.1016/j.neuron.2019.07.014 | Multiple | See *Supplementary file 5* |
| Recombinant DNA reagent | CRISPR KO (plasmid) | This paper, adapted from Addgene_52961 | | |
| Recombinant DNA reagent | Dual guide sgRNA library (plasmid library) | *Replogle et al., 2022a*; https://doi.org/10.7554/eLife.81856 | | |

## iPSC culture and neuron differentiation

Procurement and use of WTC11 cells from the Coriell cell repository followed all policies of the NIH Intramural research program. WTC11 cells were derived from an apparently healthy 30-year-old male. iPSC culture was performed following previously described protocols (*Fernandopulle et al., 2018*). Briefly, iPSCs were grown in Essential 8 Medium (Thermo Fisher Scientific, A1517001) on tissue culture plates that had been coated with Matrigel (Corning, 354277). Medium was replaced daily or as needed. Accutase (Thermo Fisher Scientific, A1110501) was used to dissociate cells for passaging as single cells. 50 nM Chroman-1 (MedChemExpress, HY-15392) was added to promote survival after thaw and accutase treatment, until cell colonies had expanded to at least 4–5 cells/colony. The iPSCs used here were previously engineered to express mouse neurogenin-2 (mNGN2) with a doxycycline-inducible promoter for neuronal differentiation (i11w-mN) (*Wang et al., 2017*). Many experiments used a line that additionally expressed a catalytically inactive dCas9 fused to a KRAB transcriptional repression domain for CRISPRi knockdowns (*Tian et al., 2019*). These engineered cells containing dCas9 are referred to here as i11w-mNC iPSCs.

iNeuron differentiation and culture was performed following previously described protocols (*Fernandopulle et al., 2018*). Briefly, on day 0 of neuronal differentiation, iPSCs were passaged with accutase and plated on Matrigel-coated tissue culture plates. iPSCs were plated in induction media (KO DMEM/F12 media [Life Technologies Corporation, #12660012] supplemented with N2 supplement [Life Technologies Corporation, 17502048], 1x GlutaMAX [Thermo Fisher Scientific, 35050061], 1x MEM nonessential amino acids [NEAA] [Thermo Fisher Scientific, 11140050], 50 nM Chroman-1, and 2 µg/mL doxycycline [Clontech, 631311]). Media was changed daily. On day 3 of neuronal differentiation, cells were passaged with accutase and plated on tissue culture plates that were coated with poly-L-ornithine (Sigma-Aldrich, P3655). Cells were plated in maturation media (BrainPhys media (STEMCELL Technologies, 05790) supplemented with B27 Plus Supplement (Thermo Fisher Scientific, A3582801), 10 ng/mL BDNF (PeproTech, 450-02), 10 ng/mL NT-3 (PeproTech, 450-03), 1 µg/mL mouse laminin (Sigma, L2020-1MG), 50 nM Chroman-1 (MedChem Express, HY-15392), and 2 µg/mL doxycycline (Clontech, 631311)). Half-media changes were performed every other day or every 3 days. Unless otherwise specified, all experiments were performed on day 7 neurons.

## HaloTag knock-in

1 million iPSCs were transfected with premixed 20 µg Alt-R A.s. Cas12a (Cpf1) V3 (IDT # 1081068), 200 pmol crRNA 5′-GGAAAAGTAAAAGATGTCTGAAT (IDT), and 10 µg knock-in (KI) plasmid donor (gene-synthesized and cloned into pUC57 by Genscript; synthesized sequence contains HaloTag coding sequence inserted immediately downstream TDP-43 start codon ATG and flanked by 400 bp left and right homology arms) using Nucleofector 4D with buffer P3 and program CA-137 (Lonza) and then plated onto one well of rhLamin521 (Thermo) coated six-well plate with StemFlex, RevitaCell (Thermo), and HDR enhancer V2 (IDT). Transfected cells were cultured in a 32°C incubator for 3 days before moving to 37°C incubator, and fresh medium was changed daily after transfection. One week after transfection, the iPSCs were live stained with Oregon Green (Promega #G2801) following the manufacturer's protocol. Positive cells were sorted into 96-well Matrigel-coated plates at 1 cell/well with 100 µL/well StemFlex and 1x CloneR2 (Stem Cell Technologies). 10–12 days later, single-cell clones were picked and KI clones were confirmed by 5′-KI and 3′-KI junction genomic PCR (5KI-F: 5′-ATCGACTGGGACCTATCACG, 5KI-R: 5′-GCGTACCCTCGATAAAAACG; 3KI-F: 5′- GAGACCTTCCAGGCCTTCC, 3KI-R: 5′- TCTACAATCCCC AGTTTCCA). Heterozygous KI clones were further identified by the presence of both non-KI and KI alleles using primers recognizing homology arms (WT-F: 5′- GACGCATCATAAGCCTTCAG, WT-R: 5′- CCAT GGATGAGACACACACC). Both non-KI and KI alleles were subject to Sanger sequencing to exclude heterozygous KI clones with indels in non-KI alleles.

## Halo-TDP-43 CRISPRi FACS screen

Approximately 50 million Halo-TDP-43 iPSCs were transduced with a whole-genome dual guide library (*Replogle et al., 2022a*) to a titer of approximately 70% in E8 with Chroman 1. The next day, the media was changed to E8 without Chroman 1. The next day, iPSCs were split onto new Matrigel-coated plates in E8 with chroman 1. The day following the split, iPSCs were treated with 8 µg/mL puromycin in E8 Chroman 1 to select for cells expressing an sgRNA. After selection, cells were split and differentiated into neurons following the standard protocol above. Some cells were maintained as iPSCs for the iPSC

screen. On day 3 post dox induction, neurons were split onto final PLO-coated plates. On day 7 post dox induction, cells were split for FACS. Cells were stained with 200 µg/mL Halo Ligand 646 (Promega HT1060) for 15 min in appropriate media. Papain solution (Worthington LK003176) was prepared by adding 20 mL TrypLE without phenol red (Gibco 12604-013) to one vial of papain and placing it at 37°C for 10 min. Neurons were washed with PBS and then washed twice with PBS with 0.5 mM EDTA. Papain/TrypLE was added to cells for 1–2 min at 37°C. Papain was removed without disturbing the neuron sheet, and trituration solution (Neuron culture media, 10 µM ROCK inhibitor, and one vial of DNAse freshly dissolved) was added to cells before transferring them to a conical tube. Cells were pipetted 3–10 times in the conical tube until cell clumps were gone. PBS was added to cells, and cells were pelleted by centrifugation at 200×*g* for 5 min. Neurons were resuspended in 1 mL Papain Inhibitor Media (27 mL trituration media with 3 mL Papain inhibitor) and incubated for 5–10 min. Cells were again pelleted by centrifugation at 200×*g* for 5 min and resuspended in cell culture media before transferring cell suspension into a 5 mL collection 12×75 mm polypropylene FACS tube with 35–40 µm strainer cap. Cells were kept on ice until FACS was performed. Cells were sorted using the MoFlo Astrios cell sorter (Beckman Coulter), with Summit software (Beckman Coulter) to set up a sorting template. GFP-positive cells were detected using the 488 nm laser excitation and 520/30 nm filter for capturing signal emission. Halo was detected using 640 nm laser excitation and 675/30 nm filter for emission. The cells were first gated on GFP expression, which served as a highly expressing co-expression marker for sgRNA, and then gated on Halo expression, as follows. Four groups were collected from the FACS session: the bottom 25% of iPSCs expressing Halo-TDP-43, the top 25% of iPSCs expressing Halo-TDP-43, the bottom 25% of neurons expressing Halo-TDP-43, and the top 25% of neurons expressing Halo-TDP-43. Genomic DNA was collected from each fraction using the NucleoSpin Blood Kit (Macherey Nagel 740950.50) following the manufacturer's instructions. Briefly, cells were washed in PBS and resuspended in PBS containing proteinase K. Next, BQ1 was added to each tube and mixed by vortexing. Samples were incubated at 56°C for 15 min and cooled to room temperature for at least 30 min. 100% ethanol was added to the tube and mixed by vortexing before loading onto the column. Columns were spun for 3 min at 4000×*g*, and flow-through was discarded. BQ2 was added to the column, and columns spun for 2 min at 4000×*g*. BQ2 step was repeated a total of two times. To elute genomic DNA, preheated BE was added to each column, incubated for 5 min at room temperature, and spun for 4 min at 4000×*g* in a new tube. Elution was repeated a second time. Next, sgRNA amplicons were enriched using primers containing indexing barcodes and Illumina adapters. 10 µg of genomic DNA was used per PCR reaction. PCR reactions also contained 100 µM of one of the following indexed forward primer (caagcagaagacggcatacgagatcgctcagttctgctat gctgtttccagcttagctcttaaac, caagcagaagacggcatacgagattatctgaccttgctatgctgtttccagcttagctcttaaac, caagcagaagacggcatacgagatatatgagacgtgctatgctgtttccagcttagctcttaaac, caagcagaagacggcatacgagat cttatggaattgctatgctgtttccagcttagctcttaaac), 100 µM of one of the following indexed reverse primer (aatgatacggcgaccaccgagatctacactcgtgggagcgagcacaaaaggaaactcaccctaactgt, aatgatacggcgaccaccga gatctacacctacaagataagcacaaaaggaaactcaccctaactgt, aatgatacggcgaccaccgagatctacactatagtagcta gcacaaaaggaaactcaccctaactgt, aatgatacggcgaccaccgagatctacactgcctggtggagcacaaaaggaaactcaccc taactgt) and 1x NEBNext Ultra Q5 MasterMix (NEB M0544L). Cycling parameters were as follows: 98°C for 30 s, 22–24 cycles of 98°C for 10 s, 66°C for 75 s, then a 5 min hold at 72°C after cycles and a hold at 4°C. PCR reactions were pooled and cleaned up with a PCR purification. A 0.5–0.65X SPRI cleanup was performed prior to sequencing. Half volume of SPRI beads (Beckman Coulter B23318) was added to the pooled PCR reaction and incubated at room temperature for 10 min. Beads were removed with a magnet and supernatant collected. Next, a 0.65 volume of SPRI beads was added to cleared supernatant, incubated at room temperature for 10 min, and tubes placed on a magnetic stand. Supernatant was removed and discarded. Beads were washed with fresh 80% ethanol twice and beads air-dried for 2–5 min. Sample was eluted in 30 µL EB buffer. The SRPI PCR cleanup was repeated in its entirety to remove excess primer contamination. PCR product yield, size, and quality were checked on a Bioanalyzer (Agilent) with a High Sensitivity DNA chip (Agilent B23318). Enriched PCR product is 562 base pairs in length. Samples were sequenced on the MiSeq with 1% PhiX through paired-end 50 sequencing with custom primers (Read1: gtgtgttttgagactataagtatcccttggagaaccacct tgttgg, Read2: tgctatgctgtttccagcttagctcttaaac, i7 Indexing primer: aagagctaagctggaaacagcatagca). Sequencing files were demultiplexed based on the sample PCR. To remove recombined dual guides, fastq files were filtered for reads containing paired guides for the same target gene using the parser.

py file (https://github.com/jhawrot/TDP_BORC, copy archived at *Hawrot, 2025*). The parsed dual hits file was analyzed using the standard MAGeCKFlute robust ranked algorithm (RRA) pipeline (*Wang et al., 2019*) comparing the high 25% to low 25% of the TDP-43 HaloTag fluorescence (results in *Supplementary files 1–4*). The RRA uses a negative binomial model to generate p-values to rank guides from negative to positive. Guide output files were filtered to remove low count guides and used to generate rank plots of target guides and NT negative controls. All data and code for plots is available on https://github.com/jhawrot/TDP_BORC, copy archived at *Hawrot, 2025*.

## CRISPRi meta-analysis

CRISPRi FACS screens in i[3]Neurons (*Tian et al., 2021*) were obtained from CRISPRBrain (https://crisprbrain.org/) and the Kampmann lab and underwent data harmonization with the screens presented here. First, screen phenotype scores and log fold changes were aligned by converting effect estimates to standardized beta values represented by z-scores. Next, extreme p-values were stabilized by truncation to prevent distortions in the statistical tests due to outliers or very small probabilities, ensuring a robust meta-analysis, accommodating the floating point of the system and acknowledging that this truncation will cause results to be more conservative. Standard errors of the effect estimates were derived by dividing the beta estimate by the absolute value of the z-scores. Fixed effect meta-analyses were then carried out for all possible screen combinations (combining already published versus novel screens). These functions calculate pooled estimates of effect sizes, variance, and test for heterogeneity among screens. Novelty was assessed by filtering associations that were not significant after multiple test corrections in either of the constituent datasets but were significant after false discovery rate (FDR) adjustment of the meta-analysis. Data was visualized using modified MAGeCKFlute pipelines to assess putative biological functions of significant genes using published enrichment functions including clusterProfiler, GOstats, enrich.HGT, and GSEA packages. TDP-43 expression hits were grouped into positive and negative modifiers using log2FC values, and statistical significance was assessed by gene ratio or FDR depending on the enrichment analysis. Code is available on https://github.com/NIH-CARD/V_Ryan_TDP43_meta_2024 (copy archived at *Nalls and Weller, 2024*).

## Lentivirus production and transduction

Individual CRISPRi knockdowns were performed by cloning sgRNAs into pU6-sgRNA EF1Alpha-puro-T2A-BFP (Addgene plasmid # 60955). sgRNA sequences for screen validation are listed in *Supplementary file 5*. To make lentivirus, Lenti-X human embryonic kidney (HEK) cells were split with trypsin, plated on poly-L-ornithine-coated plates, and allowed to recover. Cells were then transfected with sgRNA plasmids on a six-well plate using 3.75 µL Lipofectamine 3000 Reagent and 5 µL P3000 Enhancer Reagent (Invitrogen, L3000075), Opti-MEM (Gibco, 31985062), psPAX2 F46, pMD2G F45, and pAdVantage F44. Transfected cells were cultured for 2 days in DMEM supplemented with GlutaMAX-I (Thermo Fisher Scientific, 35050061) and 10% fetal bovine serum (Sigma-Aldrich, F4135). Media was then changed and supplemented with (1:500) viral boost reagent (ALSTEM, VB100), and cells were cultured for 2 days before media was harvested and Lenti-X Concentrator (Takara Bio USA, 631231) was added at (1:3) to concentrate the virus for 2–3 days at 4°C. Virus was then collected, aliquoted, and stored at –80°C until use.

iPSCs were transduced at scales of 0.1-1e6 cells per 100 µL concentrated virus while cells were in suspension. Cells were allowed to recover for 2 days before 10 µg/mL puromycin dihydrochloride (Sigma-Aldrich, P9620) was added for selection. For cells that were differentiated, transduction was performed 4 days before initiating neuronal differentiation.

## Validation of hits from genome-wide screens

Hits from the screen were validated by microscopy with individual sgRNA knockdowns in i11w-hT neurons. Cells were plated, transduced, and differentiated as described above. Fluorescent HaloTag ligand JF646 (Promega, GA112A) was added at 200 µg/mL approximately 15 min before microscopy. Cells were imaged on Nikon Ti2 with automated stage at 20x with exposure times of 100 ms for all channels. Laser power was as follows: 640 nm at 100%, 405 nm at 25%, and 561 nm at 100%. Halo-TDP-43 fluorescent intensity was quantified using CellProfiler (*Carpenter et al., 2006*) or Nikon Elements General Analysis 3 and graphed in GraphPad Prism.

sgRNAs for genes selected for further validation on untagged TDP-43 were transduced into i11w-mNC iPSCs and neurons differentiated as above. On day 7, neurons were fixed with 4% PFA (Thermo Fisher Scientific, 28906) for 10 min at room temperature. PFA was removed and cells washed in PBS. Cells were then permeabilized with 0.1% Triton X-100 (Sigma-Aldrich, 93443) for 10 min before blocking in 2% bovine serum albumin (BSA) (Jackson ImmunoResearch, 001-000-162) for 10 min. TDP-43 primary antibody (Proteintech, 10782-2-AP) was diluted 1:1000 in 2% BSA and added to cells at 4°C overnight. After washing, Donkey Anti-Rabbit IgG (H+L), Highly Cross-Adsorbed secondary antibody (CF488, Biotium, 20015) was added at 1:1000 in 2% BSA for 1 hr at room temperature. Cells were washed and DRAQ5 nuclear stain (Thermo Fisher Scientific, 62251) was added at 1:1000 dilution in PBS for 5 min. DRAQ5 was washed out with PBS three times before imaging. Stained cells were imaged on Nikon Ti2 with automated stage at 20x and an exposure time of 100 ms for 640 nm and 488 nm and 200 ms for 405 nm. Laser power was 100% for all channels. TDP-43 fluorescence intensity was quantified using Nikon Elements and graphed in GraphPad Prism.

## Lysosome drug treatment

i11w-hT iPSCs were cultured and differentiated into neurons. i11w-mNC, treated for 24 hr with DMSO, were used as a Halo-negative control. Microscopy was performed on day 7 of neuronal differentiation. Wells were individually treated with drugs by removing half the media and adding back an equal volume of media containing 2x final concentration of each drug, for time points as stated below. A volume of DMSO equal to the largest volume of drug was added at the earliest time point to wells as a negative control for treatment. Treatments were as follows: 50 µM Chloroquine (Selleck Chemicals, S6999) for 24 hr; 50 µM CA-074me (Selleck Chemicals, S7420), a Cathepsin B Inhibitor (CTSBI), for 6 hr; 30 mM Ammonium Chloride (NH$_4$Cl) (Sigma-Aldrich, A9434) for 5 hr; 500 nM Bafilomycin A1 (BafA1) (LKT Labs, B0025) for 4 hr; 1x Lysosome Protease Inhibitor Cocktail (20 µM E64 [Selleck Chemicals, S7379], 10 µM pepstatin A [Selleck Chemicals, S7381] and 50 µM leupeptin [Sigma-Aldrich, L2884]; referred to in Figure S5B-C as Protease Inhibitor) for 4 hr; 1 mM L-Leucyl-L-Leucine methyl ester (hydrochloride) (LLOME) (Cayman Chemical Company, 16008) for 4 hr; 400 µM Gly-Phe-β-naphthylamide (GPN) (Cayman Chemical Company, 14634) for 1 hr; 500 µM Sodium (meta) arsenite (NaAsO$_2$) (Sigma-Aldrich, S7400), 1 hr; 40 µM Anisomycin (Sigma-Aldrich, A5862) for 1 hr; and 0.1 mg/mL Cycloheximide solution (CHX) (Sigma-Aldrich, C4859) for 45 min. Fluorescent HaloTag ligand was added as described above 15 min prior to imaging. Cells were imaged on Nikon Ti2 with automated stage at 20x in 640 nm with an exposure time of 100 ms and 100% laser power. Halo-TDP-43 fluorescent intensity was quantified using Nikon Elements and graphed in GraphPad Prism.

## Live-cell imaging of TDP-43 motion and organelle co-localization

i11w-hT iPSCs were transduced with fluorescent tags for mitochondrial targeted mEmerald and LAMP1 tagged with mApple. The cells were then transduced with either BORCS6 or BORCS7 knockdown and NT control guides. BORCS6 knockdown cells were cultured and differentiated as described above. BORCS7 knockdown cells were differentiated as neurospheres. Fluorescent HaloTag ligand was added to day 7 BORCS6 knockdown neurons and day 10 BORCS7 knockdown neurospheres, as described above. Cells were imaged on Nikon Ti2 with automated stage at 60x and exposure times of 100 ms for all channels. Laser power was 100% for 640 nm and 561 nm. 405 nm and 488 nm lasers were set at 100% and 99.4%, respectively, for BORCS6 knockdown cells, and at 50% for BORCS7 knockdown neurospheres. DAPI image was taken once at the beginning of the video; GFP, Cy5, and RFP channels were captured over 30 s with no delay between frames and an approximate frames per second of 1.16. Puncta were counted from kymographs of axonal segments made in ImageJ or Nikon Elements and graphed in GraphPad Prism.

## qPCR

qPCR was performed using TaqMan Real-Time PCR assays. RNA was extracted from BORC and TDP-43 knockdown and NT control i11-mNC-derived neurons using the RNeasy Plus Mini kit (QIAGEN, 74134). RNA concentrations were measured with a DeNovix DS-11 FX+1 µL Spectrophotometer, and 855 ng of RNA was used as input for reverse transcription. Reverse transcription was performed using the High-Capacity cDNA Reverse Transcription Kit (Thermo Fisher Scientific, 4368814). The qPCR reaction was performed with 2x TaqMan Universal PCR Master Mix (Thermo Fisher Scientific, 4304437)

on an Applied Biosystems QuantStudio 6 Flex Real-Time PCR System. Probes were run singleplex. The Taqman probe for the target was TARDBP (FAM-MGB Hs00606522_m1, Thermo Fisher Scientific, 4453320). Two endogenous controls were used: PPIA (VIC-MGB-PL Hs04194521_s1, Thermo Fisher Scientific, 4448484) and PGK1, TaqMan ID: Hs00943178_g1, VIC-MGB-PL (Thermo Fisher Scientific, 4448484). Relative change in gene expression was examined using the $2^{-(\Delta\Delta Ct)}$ analysis method (*Livak and Schmittgen, 2001*). Significance was evaluated using an ordinary one-way ANOVA with the Šidák correction for multiple comparisons.

## pLentiCRISPR

An NT (AGAGTTACGTCGCTTCGATC) and two sgRNA targeting sequences against BORCS7 (1: CGCGATTACGTCAGTACCAC, 2: GATTACGTCAGTACCACAGG) were cloned into the lenti-CRISPR v2 plasmid (Addgene 52961) following the Zhang Lab protocol (https://media.addgene.org/data/plasmids/52/52961/52961-attachment_B3xTwla0bkYD.pdf). Briefly, the plasmid was digested with BsmBI-v2 (NEB R0739), the cut vector was dephosphorylated with NEB Quick CIP (NEB M0525), and digested and dephosphorylated vector was extracted from an agarose gel. Primers containing the sgRNA targeting sequences (Eurofins) were phosphorylated with T4 PNK (NEB M0201) and annealed before being ligated into the cut vector using 2x Ligation Master Mix (Takara 6023). After sequencing to confirm correct cloning, plasmids were transfected into Lenti-X HEK cells using standard protocols (see above). Virus was then transduced into i11w-mN iPSCs and selected with puromycin using standard protocols (see above). Selected iPSCs were differentiated using standard protocols (see above). On day 7 post dox induction, neurons were fixed and immunofluorescence performed as above.

## SunTag TDP-43 translation assay

TDP-43 coding sequence and 3'UTR were cloned into a plasmid expressing 24 SunTag repeats under an Ef1alpha promoter followed by 24 PP7 stem loops (*Ruijtenberg et al., 2018*) (modified from Addgene 74928). This plasmid was transfected into iPSCs stably expressing the SunTag scFv tagged with HaloTag under an Ef1alpha promoter (modified from Addgene 60907), PP7 coat protein tagged to 3 SnapTags (synthesized), and mEmerald-tagged lysosomes (to validate BORC KD). iPSCs were transduced with sgRNA and differentiated as spheres before plating on an eight-well glass bottom μ-Slide (Ibidi, 80827). Neurospheres were imaged at d10 post dox induction on a Nikon Ti2 spinning disk confocal microscope at 60x with only neurites in the field of view. Exposure times were as follows: 300 ms for 640 nm, 200 ms for 561, 200 ms for 488, and 100 ms for 405. Laser power was at 50% for DAPI and 100% for all other channels. Images were quantified for co-localized PP7 and SunTag (SnapTag and HaloTag, respectively) puncta using Nikon Elements. Resulting data were graphed in GraphPad Prism.

## RNA sequencing

RNA was extracted from day 7 BORCS6 knockdown and NT control i11-mNC neurons using the Direct-zol-96 MagBead RNA kit (Zymo Research, R2103). The manufacturer's protocol was followed, including the optional DNAase step, with volumes increased for an automated run on the KingFisher. Quantified with Qbit and frozen at –80°C. RNA quality was checked on a BioAnalyzer and Strand-Specific RNA-seq library preparation was performed with rRNA depletion using the NEBNext Ultra II Directional RNA Library Prep Kit for Illumina (New England Biolabs, E7760) and the NEBNext Globin & rRNA Depletion Kit (New England Biolabs, E7750). Samples were run on an Illumina NovaSeq X 10B on a 2×150 bp flow cell.

## Bulk RNA sequencing analysis

Gene length normalized counts per gene were generated and then within-sample normalization using counts per million was performed. Next, pedestalling using a value of 2 and log2 transformation was performed. Cyclic loess was used to normalize expression across samples followed by exploratory analysis to inspect for and remove outlier samples if present. If present, cyclic loess normalization is repeated. Data post cyclic loess normalization is then noise modeled (CV~mean) by experiment condition (NT v KD) and a noise threshold for the data defined. This noise threshold is then used to filter and remove noise-biased genes, with surviving genes then subject to differential testing using the appropriate test. After differential testing, all statistics are captured and exported along

with confirmatory visual generation post differential gene selection. Pathway analysis was performed to identify enriched pathways and functions for each comparison performed as part of differential testing. Alternative polyadenylation was assessed using REPAC (*Imada et al., 2023*). Results that had an absolute compositional fold change (cFC) ≥ 0.25 and an FDR-corrected p-value<0.05 were considered polyA sites that had a significant shortening (negative cFC) or lengthening event (positive cFC) and were plotted using VolcaNoseR (*Goedhart and Luijsterburg, 2020*). GO analysis of alternative polyA was performed using ShinyGO (*Ge et al., 2020*).

## Cryptic splicing analysis

For cryptic splicing analyses, sequencing files were aligned to GRCh38 reference genome (STAR v2.7.3a) (*Dobin et al., 2013*). We ran MAJIQ (v2.1) (*Vaquero-Garcia et al., 2016*) on aligned BAM files using a custom Snakemake (v 5.5.4) (*Mölder et al., 2021*) pipeline (https://github.com/frattalab/splicing; *Brown, 2023*). A threshold of 10% difference in percent spliced in (PSI, $\Delta \Psi$) was used to identify a significant change between groups. We defined cryptic splicing as junctions with $\Delta \Psi > 10\%$ that were present in <5% of control samples or $\Delta \Psi < -10\%$ that were present in >10% of control samples. An additional pipeline developed previously (*Seddighi et al., 2024*) was used to visualize and categorize each mis-spliced junction as cryptic exon, exon skipping, intron retention, or canonical junction (https://github.com/NIH-CARD/proteogenomic-pipeline; *syedislamuddin, 2024*).

## Proteomic sample preparation

BORC knockdown and NT control i11w-mNC cells were cultured and differentiated as described above until day 5 of neuronal differentiation. On day 5, half of the cultures were changed to heavy-SILAC media ('dSILAC samples'), which was DMEM for SILAC (Thermo Fisher Scientific, 88364) supplemented with B27 Plus Supplement (Thermo Fisher Scientific, A3582801), 10 ng/mL BDNF (PeproTech, 450-02), 10 ng/mL NT-3 (PeproTech, 450-03), 1 µg/mL mouse laminin (Sigma, L2020-1MG), 50 nM Chroman 1 (MedChemExpress Catalog #HY-15392), 2 µg/mL doxycycline (Clontech, 631311), 0.5 mM lysine, and 0.3 mM arginine. Heavy SILAC media was supplemented with L-Lysine:2HCl (13C6, 99%; 15N2, 99%) (Cambridge Isotope Laboratories, Inc, CNLM-291-H) and L-Arginine:HCl (13C6, 99%; 15N4, 99%) (Cambridge Isotope Laboratories, Inc, CNLM-539-H). Remaining cultures were maintained in light SILAC media made with L-Lysine monohydrochloride (SIGMA, L8662 and L7039) and L-Arginine monohydrochloride (SIGMA, A4599) to produce non-SILAC samples to evaluate changes in total protein abundance. Half-media changes were performed every other day or every 3 days. Cells were harvested on day 7 and day 15 post dox induction.

Protein was extracted from day 7 and day 15 neurons for mass spectrometry as done previously (*Reilly et al., 2023*). Neurons were gently washed with PBS, then SP3 lysis buffer (50 mM Tris-HCl [Thermo Fisher Scientific, 15568025], 50 mM NaCl [Thermo Fisher Scientific, 24740011], 1% SDS [Thermo Fisher Scientific, 15553027], 1% Triton X-100 [MilliporeSigma, X100], 1% NP-40 [Thermo Fisher Scientific, 85124], 1% Tween 20 [MilliporeSigma, P9416], 1% glycerol [MP Biomedicals, 800687], 1% Sodium deoxycholate [wt/vol] [MilliporeSigma D6750], 5 mM EDTA, 5 mM Dithiothreitol [DTT] [Thermo Fisher Scientific, R0862], 5 KU Benzonase [Sigma-Aldrich, E8263], and 1×cOmplete ULTRA Protease Inhibitor Tablets [one tablet per 10 mL of lysis buffer] [Sigma-Aldrich, 5892791001]) was added directly to the wells. Neurons were scraped into the buffer to aid with lysis. Proteins were alkylated with 10 mM iodoacetamide (Thermo Fisher Scientific, A39271) at room temperature in the dark, then reduced by adding 2 µL of 500 mM DTT to the lysate and incubating at 25°C for 30 min in the light. A Thermo Fisher KingFisher Apex was used to perform the automated SP3 protein enrichment protocol. This protocol uses SP3 beads: GE SpeedBeads (GE45152105050250) and of GE SpeedBeads (GE65152105050250) combined in a 1:1 ratio. The SP3 beads are washed in LC-MS grade water (Pierce, 51140) and reconstituted in 50% EtOH at a concentration of 4 mg/mL. Samples are diluted 1:1 with 100% EtOH, washed in 80% EtOH, and eluted in 50 mM ammonium bicarbonate with 20 µg/mL lyophilized trypsin/LysC (Promega, V5073). After completion of the KingFisher SP3 protocol, samples were shaken overnight at 37°C. The KingFisher was used to perform a bead removal protocol twice. Solvents were removed in a speed vac. Samples were stored dry at –20°C until resuspension in 2% acetonitrile (Thermo Fisher Scientific, 51101), 0.4% trifluoroacetic acid (Thermo Fisher Scientific, 85183), and 0.1% formic acid (Thermo Fisher Scientific, 28905).

## LC-MS analysis

Peptide concentration from each sample was measured on a DeNovix DS-11 FX+1 µL Spectrophotometer and normalized to 0.5 µg/µL. LC-MS/MS experiments were performed on a Thermo Scientific Orbitrap Lumos Tribrid Mass Spectrometer coupled with a Thermo Scientific Ultimate 3000 HPLC system. The injection volume for each sample was 2 µL. Peptides were separated on a Thermo Scientific ES902 column (particle size: 2 µm, diameter: 75 µm, length: 25 cm). Mobile phase A contains 0.1% formic acid in LC-MS grade water, and Mobile phase B contains 0.1% formic acid in LC-MS grade acetonitrile. Mobile phase B was increased from 3% to 22% in 64 min, then from 22% to 36% in 8 min. LC-MS/MS data were acquired in data-dependent mode. The MS1 scans were performed in orbitrap with a resolution of 120 K, a mass range of 375–1500 m/z, and an AGC target of $4\times10^5$. The quadrupole isolation window is 1.6 m/z. The precursor ion intensity threshold to trigger the MS/MS scan was set at $1\times10^4$. MS2 scans were conducted in the ion trap. Peptides were fragmented with the HCD method, and the collision energy was set at 30%. The maximum ion injection time was 35 ms. MS1 scan was performed every 3 s. As many MS2 scans were acquired within the 3 s cycle.

## Proteomic data analysis

Proteomic raw data were analyzed using MaxQuant software (v. 1.6.14) (*Cox and Mann, 2008*). A reviewed Swiss-Prot *Homo sapiens* database and a custom neuron-specific contaminant FASTA library (*Frankenfield et al., 2022*) (available to download here; *Hao Group, 2024*) were used for protein and peptide identifications with a 1% FDR cutoff. A maximum of two missed cleavages was allowed. A maximum of three variable modifications was allowed for cysteine *N*-ethylmaleimide, methionine oxidation, and protein N-terminus acetylation. Precursor and fragment tolerances were set to 25 ppm. Contaminant proteins and peptides were removed. Protein intensities from protein.txt output files were used for non-SILAC data analysis. dSILAC data was analyzed using a previously established workflow (*Hasan et al., 2023*). A standard multiplicity with two labels was enabled for SILAC data, and Lys8/Arg10 was selected as the heavy channel. For SILAC proteomics, only y-type product ions with up to two charges were used for quantification. The 'peptide.txt' report was exported for subsequent analysis in R. Peptides from contaminant proteins and peptides with intensities below 1000 were removed. Peptides with heavy/light ratios lower than 0.01 or higher than 100 were also removed to avoid outlier half-life measurements. A single time point calculation was used to calculate peptide half-lives assuming a steady state of turnover: $t_{1/2} = t_s \times \frac{ln(2)}{ln(1+R)}$, where R is the heavy-to-light ratio and $t_s$ is the labeling time point after media switch as described previously (*Hasan et al., 2023*). Protein half-lives were then determined as the harmonic mean of half-lives from unique peptides belonging to each protein.

Phosphopeptides were identified in cells cultured exclusively in light media. We employed the FragPipe LFQ-phospho workflow to detect and quantify modified peptides from our spectral files. This workflow integrates MSFragger for peptide identification with match-between-runs enabled (*Kong et al., 2017*), PTMProphet for precise mass modification localization (*Shteynberg et al., 2019*), and Philosopher for FDR filtering and reporting (*da Veiga Leprevost et al., 2020*). Differential expression of phosphopeptides was subsequently assessed in R (version 4.4.3) using the limma package (version 3.62.2). A threshold of a 1 log-fold change combined with a Benjamini-Hochberg adjusted p-value of 0.05 was applied to define significantly differentially expressed phosphosites.

## Materials availability

Cell lines will be made available to researchers by contacting wardme@nih.gov. Plasmids will be made available to researchers by contacting the corresponding authors.

## Acknowledgements

We thank Sami Barmada, Derek Narendra, and Achim Werner for their insights into the best genes from the screens to follow up on within their subject matter expertise. We thank the NIH Intramural Sequencing Center for performing RNA sequencing, the Sequencing Cores at the National Heart, Lung, and Blood Institute for sequencing screen samples, and the Proteomics Core at the National Institute of Neurological Disorders and Stroke for performing mass spec. This work was supported by the National Institute of Neurological Disorders and Stroke and the Center for Alzheimer's and Related

Dementias, within the Intramural Research Program of the National Institute on Aging, National Institutes of Health, Department of Health and Human Services. MEW was additionally supported by funding through the Chan Zuckerberg Initiative. VHR was supported in part by the National Institute of General Medical Sciences Fi2GM142475. LH was supported in part by the National Institute of Neurological Disorders and Stroke R01NS121608.

## Additional information

### Competing interests

Faraz Faghri, Nicholas L Johnson, Mike A Nalls: Participation in this project was part of a competitive contract awarded to DataTecnica LLC by the National Institutes of Health to support open science research. The other authors declare that no competing interests exist.

### Funding

| Funder | Grant reference number | Author |
| --- | --- | --- |
| National Institute of General Medical Sciences | Fi2GM142475 | Veronica H Ryan |
| National Institute of Neurological Disorders and Stroke | R01NS121608 | Ling Hao |
| National Institute of Neurological Disorders and Stroke | Intramural Research Program | Michael Emmerson Ward |
| National Institute on Aging | Intramural Research Program | Veronica H Ryan |

The funders had no role in study design, data collection and interpretation, or the decision to submit the work for publication.

### Author contributions

Veronica H Ryan, Conceptualization, Data curation, Formal analysis, Investigation, Visualization, Writing – original draft; Sydney Lawton, Joel F Reyes, Jizhong Zou, Dragan Maric, Investigation, Writing – review and editing; James Hawrot, Data curation, Software, Formal analysis, Investigation, Visualization, Writing – review and editing; Ashley M Frankenfield, Sahba Seddighi, John Replogle, Yue Andy Qi, Hebao Yuan, Formal analysis, Investigation, Writing – review and editing; Daniel M Ramos, Formal analysis, Investigation, Methodology, Writing – review and editing; Jacob Epstein, Formal analysis, Writing – review and editing; Faraz Faghri, Data curation, Writing – review and editing; Nicholas L Johnson, Martin Kampmann, Mike A Nalls, Data curation, Formal analysis, Writing – review and editing; Kory Johnson, Data curation, Formal analysis, Visualization, Writing – review and editing; Ling Hao, Data curation, Formal analysis, Investigation, Visualization, Writing – review and editing; Michael Emmerson Ward, Conceptualization, Resources, Supervision, Investigation, Project administration, Writing – review and editing

### Author ORCIDs

Veronica H Ryan ⓘ https://orcid.org/0000-0001-5121-4672
Ashley M Frankenfield ⓘ https://orcid.org/0000-0001-9311-4194
Martin Kampmann ⓘ https://orcid.org/0000-0002-3819-7019
Yue Andy Qi ⓘ https://orcid.org/0000-0003-1914-8710
Hebao Yuan ⓘ https://orcid.org/0000-0003-4425-3678
Michael Emmerson Ward ⓘ https://orcid.org/0000-0002-5296-8051

Reviewer #1 (Public review): https://doi.org/10.7554/eLife.104057.3.sa1
Reviewer #2 (Public review): https://doi.org/10.7554/eLife.104057.3.sa2
Reviewer #3 (Public review): https://doi.org/10.7554/eLife.104057.3.sa3
Author response https://doi.org/10.7554/eLife.104057.3.sa4

## Additional files

### Supplementary files

Supplementary file 1. MAGeCKFlute robust ranked algorithm (RRA) results for genes that increased Halo-TDP-43 levels in i3Neurons.

Supplementary file 2. MAGeCKFlute robust ranked algorithm (RRA) results for genes that decreased Halo-TDP-43 levels in i3Neurons.

Supplementary file 3. MAGeCKFlute robust ranked algorithm (RRA) results for genes that increased Halo-TDP-43 levels in induced pluripotent stem cells (iPSCs).

Supplementary file 4. MAGeCKFlute robust ranked algorithm (RRA) results for genes that decreased Halo-TDP-43 levels in induced pluripotent stem cells (iPSCs).

Supplementary file 5. Individual guide sequences cloned for screen validation. Secondary screen results summaries and figures used in are indicated in the last three columns.

MDAR checklist

### Data availability

RNA sequencing data files can be accessed at GEO Accession Number GSE299976 and the Alzheimer's Disease Workbench (ADWB) https://fair.addi.ad-datainitiative.org/#/data/datasets/borcs6_kd_transcriptomics_on_ineurons). The mass spectrometry proteomics data have been deposited to the ProteomeXchange Consortium via the PRIDE partner repository with the dataset identifier PXD06320665 (*Perez-Riverol et al., 2022*).

The following datasets were generated:

| Author(s) | Year | Dataset title | Dataset URL | Database and Identifier |
|---|---|---|---|---|
| Ryan et al | 2025 | Maintenance of neuronal TDP-43 expression requires axonal lysosome transport | https://www.ncbi.nlm.nih.gov/geo/query/acc.cgi?acc=GSE299976 | NCBI Gene Expression Omnibus, GSE299976 |
| Ryan et al | 2025 | Maintenance of neuronal TDP-43 expression requires axonal lysosome transport | https://proteomecentral.proteomexchange.org/cgi/GetDataset?ID=PXD063206 | ProteomeXchange, PXD063206 |

The following previously published datasets were used:

| Author(s) | Year | Dataset title | Dataset URL | Database and Identifier |
|---|---|---|---|---|
| Tian R, Abarientos A, Hong J, Hashemi SH, Yan R, Dräger N, Leng N, Nalls MA, Singleton AB, Xu K, Faghri F, Kampmann M | 2020 | Glutamatergic Neuron-CellRox-CRISPRi | https://crisprbrain.org/simple-screen/?screen=Glutamatergic%20Neuron-CellRox-CRISPRi | CRISPRbrain, GlutamatergicNeuron-CellRox-CRISPRi |
| Tian R, Abarientos A, Hong J, Hashemi SH, Yan R, Dräger N, Leng K, Nalls MA, Singleton AB, Xu F, Faghri F, Kampmann M | 2020 | Glutamatergic Neuron-Liperfluo-CRISPRi | https://crisprbrain.org/simple-screen/?screen=Glutamatergic%20Neuron-Liperfluo-CRISPRi | CRISPRbrain, GlutamatergicNeuron-Liperfluo-CRISPRi |

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
