## [Editor Report · eLife Assessment]

In this **important** manuscript, Ryan et al perform a genome-wide CRISPR based screen to identify genes that modulate TDP-43 levels in neurons. They identify a number of genes and pathways and highlight the BORC complex, which is required for anterograde lysosome transport as one such regulator of TDP-43 protein levels. Overall, this is a **convincing** study, which opens the door for additional future investigations on the regulation of TDP-43.

---

## [Referee Report · Reviewer #1 (Public review)]

Summary:

As TDP-43 mislocalization is a hallmark of multiple neurodegenerative diseases, the authors seek to identify pathways that modulate TDP-43 levels. To do this, they use a FACS based genome wide CRISPR KD screen in a Halo tagged TDP-43 KI iPSC line. Their screen identifies a number of genetic modulators of TDP-43 expression including BORC which plays a role in lysosome transport.

Strengths:

Genome wide CRISPR based screen identifies a number of modulators of TDP-43 expression to generate hypotheses regarding RNA BP regulation and perhaps insights into disease

---

## [Referee Report · Reviewer #2 (Public review)]

Summary:

The authors employ a novel CRISPRi FACS screen and uncover the lysosomal transport complex BORC as a regulator of TDP-43 protein levels in iNeurons. They also find that BORC subunit knockouts impair lysosomal function, leading to slower protein turnover and implicating lysosomal activity in the regulation of TDP-43 levels. This is highly significant for the field given that (a) other proteins could also be regulated in this way, (b) understanding mechanisms that influence TDP-43 levels are significant given that its dysregulation is considered a major driver of several neurodegenerative diseases and (c) the novelty of the proposed mechanism.

Strengths:

The novelty and information provided by the CRISPRi screen. The authors provide evidence indicating that BORC subunit knockouts impair lysosomal function, leading to slower protein turnover and implicating lysosomal activity in the regulation of TDP-43 levels and show a mechanistic link between lysosome mislocalization and TDP-43 dysregulation. The study highlights the importance of localized lysosome activity in axons and suggests that lysosomal dysfunction could drive TDP-43 pathologies associated with neurodegenerative diseases like FTD/ALS. Further, the methods and concepts will have an impact to the larger community as well. The work also sets up for further work to understand the somewhat paradoxical findings that even though the tagged TDP-43 protein is reduced in the screen, it does not alter cryptic exon splicing and there is a longer TDP-43 half-life with BORC KD.

---

## [Referee Report · Reviewer #3 (Public review)]

Summary:

In this work, Ryan et al. have performed a state-of-the-art full genome CRISP-based screen of iNEurons expressing a teggd version of TDP-43 in order to determine expression modifiers of this protein. Unexpectedly, using this approach the authors have uncovered a previously undescribed role of the BORC complex in affecting the levels of TDP-43 protein, but not mRNA expression. Taken together, these findings represent a very solid piece of work that will certainly be important for the field.

Strengths:

BORC is a novel TDP-43 expression modifier that has never been described before and it seemingly acts on regulating protein half life rather than transcriptome level. It has been long known that different labs have reported different half-lives for TDP-43 depending on the experimental system but no work has ever explained these discrepancies. Now, the work of Ryan et al. has for the time identified one of these factors which could account for these differences and play an important role in disease (although this is left to be determined in future studies).

The genome wide CRISPR screening has demonstrated to yield novel results with high reproducibility and could eventually be used to search for expression modifiers of many other proteins involved in neurodegeneration or other diseases

---

## [Author Response]

The following is the authors’ response to the original reviews.

**Reviewer #1 (Public review):**
Summary: As TDP-43 mislocalization is a hallmark of multiple neurodegenerative diseases, the authors seek to identify pathways that modulate TDP-43 levels. To do this, they use a FACS based genome wide CRISPR KD screen in a Halo tagged TDP-43 KI iPSC line. Their screen identifies a number of genetic modulators of TDP-43 expression including BORC which plays a role in lysosome transport.Strengths:Genome wide CRISPR based screen identifies a number of modulators of TDP-43 expression to generate hypotheses regarding RNA BP regulation and perhaps insights into disease.Weaknesses:It is unclear how altering TDP-43 levels may relate to disease where TDP-43 is not altered in expression but mislocalized. This is a solid cell biology study, but the relation to disease is not clear without providing evidence of BORC alterations in disease or manipulation of BORC reversing TDP-43 pathology in disease.

We thank the reviewer for this comment and have updated the discussion to include more discussion of the role TDP-43 may play in the BORCS8-associated neurodegenerative disorder and how understanding how lysosome localization changing TDP-43 levels may help patients (lines 313-321).

The mechanisms by which BORC and lysosome transport modulate TDP-43 expression are unclear. Presumably, this may be through altered degradation of TDP protein but this is not addressed.

We agree with the reviewer that understanding the mechanism by which lysosome transport regulates TDP-43 levels is important and plan to examine this in future studies.

Previous studies have demonstrated that TDP-43 levels can be modulated by altering lysosomal degradation so the identification of lysosomal pathways is not particularly novel.

We thank the reviewer for this comment and have updated the text to make this clearer (lines 310-313). What hasn’t been observed previously is a change in lysosome localization affecting TDP-43 levels.

It is unclear whether this finding is specific to TDP-43 levels or whether lysosome localization may more broadly impact proteostasis in particular of other RNA BPs linked to disease.

We agree that this is an interesting question and something that should be investigated in future studies.

Unclear whether BORC depletion alters lysosome function or simply localization.

We thank the reviewer for this comment. Lysosome function related to protein turnover has not yet been examined in the literature after loss of BORC, but other aspects of lysosome function (including lipid metabolism and autophagic flux) have been shown to be disrupted upon loss of BORC. We have updated the discussion to address this (lines 292-296).

**Reviewer #2 (Public review):**
Summary: The authors employ a novel CRISPRi FACS screen and uncover the lysosomal transport complex BORC as a regulator of TDP-43 protein levels in iNeurons. They also find that BORC subunit knockouts impair lysosomal function, leading to slower protein turnover and implicating lysosomal activity in the regulation of TDP-43 levels. This is highly significant for the field given that (a) other proteins could also be regulated in this way, (b) understanding mechanisms that influence TDP-43 levels are significant given that its dysregulation is considered a major driver of several neurodegenerative diseases and (c) the novelty of the proposed mechanism.Strengths:The novelty and information provided by the CRISPRi screen. The authors provide evidence indicating that BORC subunit knockouts impair lysosomal function, leading to slower protein turnover and implicating lysosomal activity in the regulation of TDP-43 levels and show a mechanistic link between lysosome mislocalization and TDP-43 dysregulation. The study highlights the importance of localized lysosome activity in axons and suggests that lysosomal dysfunction could drive TDP-43 pathologies associated with neurodegenerative diseases like FTD/ALS. Further, the methods and concepts will have an impact to the larger community as well. The work also sets up for further work to understand the somewhat paradoxical findings that even though the tagged TDP-43 protein is reduced in the screen, it does not alter cryptic exon splicing and there is a longer TDP-43 half-life with BORC KD.Weaknesses:While the data is very strong, the work requires some additional clarification.

We thank the reviewer for these comments. Our detailed responses are included below in the “recommendations for authors” section.

**Reviewer #3 (Public review):**
Summary: In this work, Ryan et al. have performed a state-of-the-art full genome CRISP-based screen of iNeurons expressing a tagged version of TDP-43 in order to determine expression modifiers of this protein. Unexpectedly, using this approach the authors have uncovered a previously undescribed role of the BORC complex in affecting the levels of TDP-43 protein, but not mRNA expression. Taken together, these findings represent a very solid piece of work that will certainly be important for the field.Strengths:BORC is a novel TDP-43 expression modifier that has never been described before and it seemingly acts on regulating protein half life rather than transcriptome level. It has been long known that different labs have reported different half-lives for TDP-43 depending on the experimental system but no work has ever explained these discrepancies. Now, the work of Ryan et al. has for the time identified one of these factors which could account for these differences and play an important role in disease (although this is left to be determined in future studies).The genome wide CRISPR screening has demonstrated to yield novel results with high reproducibility and could eventually be used to search for expression modifiers of many other proteins involved in neurodegeneration or other diseasesWeaknesses:The fact that TDP-43 mRNA does not change following BORCS6 KD is based on a single qRT- PCR that does not really cover all possibilities. For example, the mRNA total levels may not change but the polyA sites may have switched from the highly efficient pA1 to the less efficient and nuclear retained pA4. There are therefore a few other experiments that could have been performed to make this conclusion more compelling, maybe also performing RNAscope experiments to make sure that no change occurred in TDP-43 mRNA localisation in cells.

We thank the reviewer for this comment. To address this point, we performed an analysis of polyA sites on our RNA sequencing data using REPAC and did not find a change in TDP-43 poly adenylation after BORC KD (Figure S6C). Other transcripts do have altered polyA sites, which are summarized in Figure S6C. We also performed HCR FISH for TARDBP mRNA in TDP-43 and BORC KD neurons. While we did not see a difference in RNA localization (see A below, numbers on brackets indicate p-values), we also were not able to detect a significant difference in total TARDBP mRNA levels upon TDP-43 KD (see B below, numbers on brackets indicate p-values), suggesting that some of the signal detected is non-specific to TARDBP. Because of this, we cannot conclusively say that BORC KD does not alter TARDBP mRNA localization using the available tools.

**Author response image 1. sa4fig1:** 

Even assuming that the mRNA does not change, no explanation for the change in TDP-43 protein half life has been proposed by the authors. This will presumably be addressed in future studies: for example, are mutants that lack different domains of TDP-43 equally affected in their half-lives by BORC KD?. Alternatively, can a mass-spec be attempted to see whether TDP-43 PTMs change following BORCS6 KD?

We agree with the reviewer that these are important experiments that could be done in the future to further examine the mechanism by which loss of BORC alters TDP-43 half-life. We examined our proteomics data for differential phosphorylation and ubiquitination in NT vs BORC KD (Figure S7G-H). We were unable to detect PTMs on TDP-43, so we cannot say if they contribute to the change in TDP-43 half-life we observed.

**Reviewer #1 (Recommendations for the authors):**
Recommendations are detailed in the public review.
**Reviewer #2 (Recommendations for the authors):**
Ryan et al, employ a CRISPRi FACS screen and uncover the lysosomal transport complex BORC as a regulator of TDP-43 protein levels in iNeurons. The authors provide strong evidence indicating that BORC subunit knockouts impair lysosomal function, leading to slower protein turnover and implicating lysosomal activity in the regulation of TDP-43 levels. The authors then provided additional evidence of TDP-43 perturbations under lysosome-inhibiting drug conditions, underscoring a mechanistic link between lysosome mislocalization and TDP-43 dysregulation. The study highlights the importance of localized lysosome activity in axons and suggests that lysosomal dysfunction could drive TDP-43 pathologies associated with neurodegenerative diseases like FTD/ALS. The work is exciting and could be highly informative for the field.Concerns: There are some disconnects between the figures and the main text that can benefit from refining of the figures to align better with the main text. This does not require additional experiments other than perhaps Figure 4B. The impact of the work could be further discussed - it is an interesting disconnect between the fact BORC KD causes decreased IF of the Halo-tagged TDP-43 and lysosomal transport, however this reduction does not impact cryptic exon expression and also increases TDP-43 half life (and of other proteins). It is a very interesting and potentially informative part of the manuscript.

We thank the reviewer for their detailed reading of our manuscript. We have endeavored to better match the figures and the text and have added more discussion of the impact of the work.

Minor:(1) Suggestion: relating to the statement "Gene editing was efficient, with almost all selected clones correctly edited." - please provide values or %.

We updated the text to remove the statement about the editing efficiency, instead saying we identified a clone that was correct for both sequence and karyotype (lines 83-85).

(2) Relating to Figure 1A: Please provide clarification regarding tagging strategy with the halotag - e.g. why in front of exon2.

We updated the figure legend to reflect that the start codon for TDP-43 is in exon 2, hence why we placed the HaloTag there.

(3) Relating to Figure S1: A and B seems to have been swapped.

We thank the reviewer for catching this mistake and have fixed the figure/text.

(4) Relating to Figure 1B: figure legend does not indicate grayscale coloring of TDP-43 signal.

We have added text in the figure legend to indicate that the Halo signal is shown in grayscale in the left-handed panels.

(5) Relating to Figure 1C: can the authors clarify abbreviation for 'NT' in text and legend.

We thank the reviewer for catching this and have indicated in the text and figure legend that NT refers to the non-targeting sgRNA that was used as a control for comparison to the TDP-43 KD sgRNA.

(6) Relating to figure 2B and S2A: main text mentioned "Non-targeting Guides" however the figure does not show non-targeting guides to confirm.

We thank the reviewer for catching this oversight, we updated the figure legends for these figures to indicate that the non-targeting (NT) guides are shown in gray on the rank plot. They cluster towards the middle, more horizontal portion of the graphs, showing that the more vertical sections of the graph are hits.

(7) Suggestion: To make it easier on the reader, please provide overlap numbers for the following statement ..."In comparing the top GO terms associated with genes that increase or decrease Halo-TDP-43 levels in iNeurons, we found that almost none altered Halo-TDP-43 levels in iPSCs...".

We thank the reviewer for this comment and have updated the text to indicate that only a single term is shared between the iPSC and iNeuron screens (lines 113-117).

(8) Relating to the statement "We cloned single sgRNA plasmids for 59 genes that either increased or decreased Halo-TDP-43 in iNeurons but not in iPSCs." Can the authors provide a list of the 59 genes.

We have included a new column in the supplemental table S1 indicating the result of the Halo microscopy validation to hopefully clarify which genes lead to a validated phenotype and which did not.

(9) Relating to the statement "To rule out the possibility of neighboring gene or off-target effects of CRISPRi, as has been reported previously15, we examined the impact of BORC knockout (KO) on TDP-43 levels. Using the pLentiCRISPR system, which expresses the sgRNA of interest on the same plasmid as an active Cas916 we found that KO of BORCS7 using two different sgRNAs decreased TDP-43 levels by immunofluorescence (Figure 5C-D)." Please provide clarification as to why BORCS7 was chosen out of all the BORCS? From the data presentation thus far (Figure 4B & 5A), the reader might have anticipated testing BORCS6 for panels 5C-D.

We thank the reviewer for this comment. We tried a couple of BORCs with the pLentiCRISPR system, but BORCS7 was the only one we were convinced we got functional knockout for based on lysosome localization. We think that either the guides were not ideal for the other BORC components we tried, or we did not get efficient gene editing across the population of cells tested. Because we had previously been working with knock down and CRISPRi guides are not the same as CRISPR knock out guides, we couldn’t use the existing guide sequences we know work well for BORC. Since loss of one BORC gene causes functional loss of the complex and restricts lysosomes to the soma, we did not feel it necessary to assay all 8 genes.

(10) Relating to the statement "We treated Halo-TDP-43 neurons with various drugs that disrupt distinct processes in the lysosome pathway and asked if Halo-TDP-43 levels changed. Chloroquine (decreases lysosomal acidity), CTSBI (inhibits cathepsin B protease), ammonium chloride (NH4Cl, inhibits lysosome-phagosome fusion), and GPN (ruptures lysosomal membranes) all consistently decreased Halo-TDP-43 levels (Figure 6A-B, S5A-C)" Please provide interpretations for Figures S5A and S5C in text.

We thank the reviewer for catching this oversight and have updated the text accordingly (lines 183-191).

(11) Relating to figure 6E: please provide in legend what the different colors used correlate with (i.e. green/brown for BORCS7 KD)?

We thank the reviewer for pointing this out. These colors were mistakenly left in the figure from a version looking to see if the observed effects were driven by a single replicate rather than a consistent change (each replicate has a slightly different color). As the colors are intermingled and not separated, we concluded the effect was not driven by a single replicate. The colors have been removed from the updated figure for simplicity.

(12) Relating to the statement "We observed a similar trend for many proteins in the proteome (Figure 8B)" This statement can benefit from stating which trend the authors are referring to, it is currently unclear from the volcano plot shown for Figure 8B.

We thank the reviewer for catching this and have updated the text accordingly.

(13) Relating to the statement "For almost every gene, we observed an increase or decrease in Halo-TDP-43 levels without a change in Halo-TDP-43 localization or compartment specific level changes (Figure 4B)." Please provide: (1) the number of genes examined, (2) additional clarification of "localization" and "compartment specific" level changes, (3) some quantification and or additional supporting data of the imaging results. Figures 5A-B presents with the same concern relating to the comment "To determine if results from Halo-TDP-43 expression assays also applied to endogenous, untagged TDP-43 levels, we selected 22 genes that passed Halo validation and performed immunofluorescence microscopy for endogenous (untagged) TDP-43 (Figure 4D-G,5A-B, S4E-F)." please clarify further.

We thank the reviewer for requesting this clarification. This statement refers to all 59 genes tested by Halo imaging; only one (MFN2) showed any hints of aggregation or changes in localization, every other gene (58) showed what appeared to be global changes in Halo-TDP-43 levels. We were initially intrigued by the MFN2 phenotype; however, we were unable to replicate it on endogenous TDP-43 and thus concluded that this might be an effect specific to the tagged protein. The representative images shown in Figure 4B are representative of the changes we observed across all 59 genes tested (if changes were present). From the 59 genes that we observed a change in Halo-TDP-43 levels by microscopy, we selected a smaller number to move forward to immunofluorescence for TDP-43. We picked a subset of genes from each of the different categories we had identified (mitochondria, m6A, ubiquitination, and some miscellaneous) to validate by immunofluorescence, thinking that genes in the same pathway would act similarly. We have added a column to the supplemental table S1 indicating which genes were tested by immunofluorescence and what the result was. We have also attempted to clarify the results section to make the above clearer.

(14) Relating to the statement "To determine if results from Halo-TDP-43 expression assays also applied to endogenous, untagged TDP-43 levels, we selected 22 genes that passed Halo validation and performed immunofluorescence microscopy for endogenous (untagged) TDP-43 (Figure 4D-G, 5A-B, S4E-F). Of these, 18 (82%) gene knockdowns showed changes in endogenous TDP-43 levels (Figure 4D-G, S4E-F)." It is difficult to identify the 18 or 22 genes in the figures as described in the main text.

We added columns to the supplemental table S1 listing the genes and the result in each assay.

(15) Relating to figures S7A and 8A and the first part of the section "TDP-43, like the proteome, shows longer turnover time in BORC KD neurons" Can the authors provide clarification why the SunTag assay was performed with BORCS6 KD (S7A) but the follow-up experiment (8A) was performed with BORCS7 KD. Does BORCS6 KD show similar results as BORCS7 with the SunTag assay, and does TDP-43 protein abundance with BORCS7 KD show similar results as BORCS6?

Because loss of any of the 8 BORC genes causes functional loss of BORC and lysosomes to be restricted to the peri-nuclear space, we used BORC KDs interchangeably. Additionally, all BORC KDs had similar effects on Halo-TDP-43 levels.

**Reviewer #3 (Recommendations for the authors):**
Adding more control experiments that TDP-43 mRNA is really not affected following BORC KD

We performed a FISH experiment to examine TARDBP mRNA localization upon BORC KD but were unable to conclusively say whether BORC KD changes TARDBP mRNA localization (see above). We also analyzed our RNA sequencing experiment for alternative polyadenylation sites upon BORC KD. Results are in Figure S6C.

Although this could be part of a future study, the authors should try and determine what are the changes to TDP-43 that drive a change in the half-life.

We agree with the reviewer that these are important experiments and hope to figure this out in the future.